# FlexRibbon: Joint Sequence and Structure Pretraining for Protein Modeling

**Jianwei Zhu**[1,*], **Yu Shi**[1,*], **Ran Bi**[1,*], **Peiran Jin**[1,*,†], **Chang Liu**[1,*,†], **Zhe Zhang**[1,*],
**Haitao Huang**[1,2], **Zekun Guo**[1], **Pipi Hu**[3], **Fusong Ju**[1], **Lin Huang**[4], **Xinwei Tai**[1,5],
**Chenao Li**[6], **Kaiyuan Gao**[5], **Xinran Wei**[1], **Huanhuan Xia**[1], **Jia Zhang**[4], **Yaosen Min**[1],
**Zun Wang**[7], **Yusong Wang**[1], **Liang He**[1], **Haiguang Liu**[1,†], **Tao Qin**[1,†]

[1]Zhongguancun Academy, Beijing, China.
[2]Hunan University, Changsha, China.
[3]Beijing Institute of Mathematical Sciences and Applications, Beijing, China.
[4]Ubiquant, Beijing, China.
[5]Huazhong University of Science and Technology, Wuhan, China.
[6]Institute of Biophysics, Chinese Academy of Sciences, Beijing, China.
[7]Shanghai Artificial Intelligence Laboratory, Shanghai, China.

## Abstract

Protein foundation models have advanced rapidly, with most approaches falling into two dominant paradigms. Sequence-based language models (e.g., ESM-2) capture sequence semantics at scale, and a number of recent works incorporate structural signals into sequence encoders. MSA-based predictors (e.g., AlphaFold 2/3) achieve accurate folding by exploiting evolutionary couplings, but their reliance on homologous sequences makes them less reliable in highly mutated or alignment-sparse regimes. We present FlexRibbon[‡], a pretrained protein model that jointly learns from amino acid sequences and three-dimensional structures. Our pretraining strategy combines masked language modeling with diffusion-based denoising, enabling bidirectional sequence-structure learning without requiring MSAs. Trained on both experimentally resolved structures and AlphaFold 2 predictions, FlexRibbon captures global folds as well as flexible conformations critical for biological function. Evaluated across diverse tasks spanning interface design, intermolecular interaction prediction, and protein function prediction, FlexRibbon establishes new state-of-the-art performance on 12 different tasks, with particularly strong gains in mutation-rich settings where MSA-based methods often struggle.

## 1 Introduction

Proteins are fundamental to nearly all biological processes, and modeling their sequences, structures, and functions underpins biomedical and biotechnological advances ranging from enzyme engineering to therapeutic antibody design. Recently, protein foundation models (PFMs) have emerged as a unifying framework that leverages large-scale data and deep learning to capture the principles of protein biology, offering new opportunities for both understanding and design. The development of PFMs has followed two main trajectories. One line of work builds on sequence-only language models (PLMs) such as ESM-2 (Lin et al., 2023) and ProtT5 (Pokharel et al., 2022), which leverage large corpora of protein sequences to learn universal embeddings. These models are broadly applicable and computationally efficient, but the lack of physical relevance, particularly information about three-dimensional geometry, limits their ability to capture the structural basis of protein function. Several subsequent efforts enrich sequence models with structural cues—such as injecting residue-level geometric features (Bepler & Berger, 2019), incorporating template-based or graph-based structural information (Heinzinger et al., 2024), distilling representations from structure predictors into sequence encoders (Ouyang-Zhang et al., 2025; Zhang et al., 2024), or perform

---

[*]These authors contributed equally.
[†]Corresponding authors: {jinpeiran, liuchang, liuhaiguang, qintao}@bza.edu.cn
[‡]Code is available at: `https://github.com/bjzgcai/FlexRibbon`

structure-conditioned sequence design (Zheng et al., 2023). These approaches improve structural awareness but remain fundamentally sequence-centric: structure is used as an auxiliary signal rather than being jointly modeled, and atomic-level geometry is not learned generatively or bidirectionally.

Another line is represented by multiple sequence alignment (MSA) based structure predictors, exemplified by AlphaFold 2/3 (Jumper et al., 2021; Abramson et al., 2024), which exploit evolutionary couplings encoded in MSAs to achieve striking accuracy in structure prediction. Yet, this dependence on homologous sequences introduces sensitivity: when alignments are shallow, sparse, or disrupted by extensive mutation, the predictive signal degrades. As a result, critical scenarios such as antibody CDR loops, intrinsically disordered interfaces, and rapidly evolving pathogens remain inadequately addressed by either paradigm; in these settings, single-sequence models that bypass MSAs and directly model individual sequences provide a more faithful way to capture flexible and highly mutated regions where alignment signals are weak.

We introduce FlexRibbon, a 3-billion-parameter pretrained protein model that learns directly from amino acid sequences and large-scale structural corpora, including experimentally resolved structures (Berman et al., 2000) and AlphaFold 2 predicted structures (Varadi et al., 2024). Unlike sequence-only models or predictors that impose a one-way sequence-to-structure mapping, FlexRibbon integrates sequence and structure signals from the outset: each residue is represented by a single embedding that combines sequence identity with structural context. The training strategy couples masked language modeling on sequences with diffusion-based denoising on structures, enabling the model to capture bidirectional sequence-structure dependencies and support full-atom structure generation. To address the variable confidence of predicted structures, we introduce an adaptive loss that selectively weights low-confidence regions, extracting useful signal while avoiding overfitting to unreliable geometry. Previous joint models were designed primarily for structure prediction, but the high memory cost of full-atom representations made the structural component difficult to scale, so most parameters ended up concentrated on the sequence side. FlexRibbon overcomes this limitation with a hierarchical modeling strategy that allocates scalable capacity across both sequence and structure, allowing efficient large-scale structural learning alongside sequence semantics.

We systematically evaluate FlexRibbon across three broad task families: (i) flexible interface prediction and design, such as antibody/nanobody CDR modeling and peptide binding; (ii) intermolecular interaction prediction, including protein-ligand docking prediction, ligand-induced conformational change, and protein-ligand affinity prediction; and (iii) protein function prediction, such as gene ontology and enzyme activity. Across these categories, FlexRibbon achieves state-of-the-art performance, with especially strong improvements in mutation-rich settings where MSA-based methods often struggle. Beyond outperforming existing models, our results highlight the consistent advantages of joint sequence-structure pretraining. The key contributions are:

- Proposing a novel pretraining strategy for FlexRibbon that unifies protein structure prediction and design by combining masked language modeling with diffusion-based denoising, thereby learning a bidirectional sequence-structure mapping rather than a one-way sequence-to-structure mapping.

- Introducing a hierarchical modeling strategy that balances scalable capacity across sequence and structure representations, overcoming the memory bottlenecks of full-atom models and enabling structural representations to scale effectively.

- Showing that FlexRibbon enables co-design of protein sequence and structure, delivering substantial improvements on flexible and highly mutated regions such as antibody/nanobody CDR loops and peptide-binding interfaces, where MSA-based models struggle.

- Demonstrating consistent gains across 12 tasks spanning flexible interface modeling, intermolecular interactions, and protein function prediction, showing that sequence-structure pretraining transfers broadly beyond protein folding.

## 2 RELATED WORKS

**Protein foundation models.** PFMs learn transferable protein representations for diverse tasks. Early PFMs were sequence-only language models such as ESM-1b/ESM-2 (Rives et al., 2021; Lin et al., 2023) and ProtT5 (Pokharel et al., 2022), trained on large sequence corpora but limited by the absence of geometric priors. A number of follow-up efforts investigates how structural knowledge

can inform sequence models, exploring ideas such as learning embeddings shaped by structural similarity (Bepler & Berger, 2019), leveraging template-based or topology-aware encoders (Heinzinger et al., 2024), transferring internal states from structure prediction networks (Ouyang-Zhang et al., 2025; Zhang et al., 2024), and designing sequences conditioned on known folds (Zheng et al., 2023). In contrast, structure-centric PFMs such as AlphaFold2/3 (Jumper et al., 2021; Abramson et al., 2024) leverage MSA and templates to achieve high-accuracy folding, yet degrade in highly mutated or low-homology regions. More recently, PFMs have moved toward multimodal, structure-aware pretraining. ESM-3 (Hayes et al., 2025) unifies sequence, structure, and function in a frontier generative model. DPLM-2 (Wang et al., 2025b) extends diffusion protein language models (PLMs) to jointly model both sequences and structures via structure tokenization.

**Antibody design.**    Antibody design methods can be broadly categorized into sequence-based and structure-based approaches. On the sequence side, general-purpose PLMs such as ProtBert (Elnaggar et al., 2021) provide strong baselines for paratope prediction, mutation recovery, and antibody library generation. More specialized pretraining frameworks such as SFM-Protein (He et al., 2024) introduce masked language modeling with pairwise and span-level objectives, showing improved performance on CDR-H3 benchmarks. In addition, graph neural network methods like ABGNN (Gao et al., 2023) and RefineGNN (Jin et al., 2022) attempt to couple sequence embeddings with local structural context, while knowledge-driven frameworks such as RosettaAntibodyDesign (Adolf-Bryfogle et al., 2018) remain widely used in practice. On the structure side, diffusion-based models such as DiffAb (Luo et al., 2022) generate CDR loops conditioned on antigen structures, enabling co-design of sequence and structure, while dyMEAN (Kong et al., 2023b) and MEAN (Kong et al., 2023a) extend this direction with $E(3)$-equivariant architectures for full-atom design. More recently, IgGM (Wang et al., 2025a) expands design capabilities to antibodies and nanobodies by producing antigen-specific complexes. Together, these approaches demonstrate the promise of combining sequence information and structural priors for flexible and functional antibody design.

## 3 METHODS

### 3.1 DIFFUSION PRETRAINING

We employ diffusion modeling as a generative pretraining objective for protein structures. A structure $\mathbf{R} \in \mathbb{R}^{3N}$ is represented by the 3D coordinates of all heavy atoms. Following Karras et al. (2022) (also adopted by AlphaFold 3), we connect the data distribution $p(\mathbf{R})$ with Gaussian noise $p_{\text{src}}$ through a variance-exploding process (Song et al., 2021):

$$\mathbf{R}_t = \mathbf{R}_0 + \sigma_t \boldsymbol{\epsilon}, \quad \boldsymbol{\epsilon} \sim \mathcal{N}(\mathbf{0}, \mathbf{I}),$$

where $\sigma_t$ increases with time $t$. Sampling amounts to reversing this process, which requires learning the score function $\nabla \log p_t$. We approximate it with a neural network $\mathbf{s}_\theta(\mathbf{R}, t)$, equivalently parameterized as:

$$\mathbf{s}_\theta(\mathbf{R}, t) := \frac{\mathbf{D}_\theta(\mathbf{R}, t) - \mathbf{R}}{\sigma_t^2}, \tag{1}$$

where $\mathbf{D}_\theta(\mathbf{R}, t)$ denotes model output. The effective learning objective amounts to denoising loss:

$$\min_\theta \mathbb{E}_{p(t)} w_t \, \mathbb{E}_{p(\mathbf{R}_0)} \mathbb{E}_{p(\mathbf{R}_t|\mathbf{R}_0)} \big\| \mathbf{D}_\theta(\mathbf{R}_t, t) - \mathbf{R}_0 \big\|^2, \tag{2}$$

with a time-step sampler $p(t)$ and weight $w_t$. A key challenge is ensuring invariance to rigid-body transformations. We remove translational freedom by centering structures on the center of mass, and enforce rotational invariance by augmenting data with random $\text{SO}(3)$ rotations, instead of relying on heavy $\text{SO}(3)$-equivariant architectures (Köhler et al., 2020) that may also introduce undesired reflection symmetry. We also found that alignment-based objectives (Xu et al., 2022; Abramson et al., 2024) did not improve training stability in our settings but added the risk of improper sampling (Wohlwend et al., 2025). Further details are provided in Appendix C.6.

### 3.2 ARCHITECTURE

Our architecture is organized in three stages: sequence module, coarse-grained structure module, and all-atom structure module (Fig. 1 and Fig. 7). This balances efficiency and expressivity: coarse-

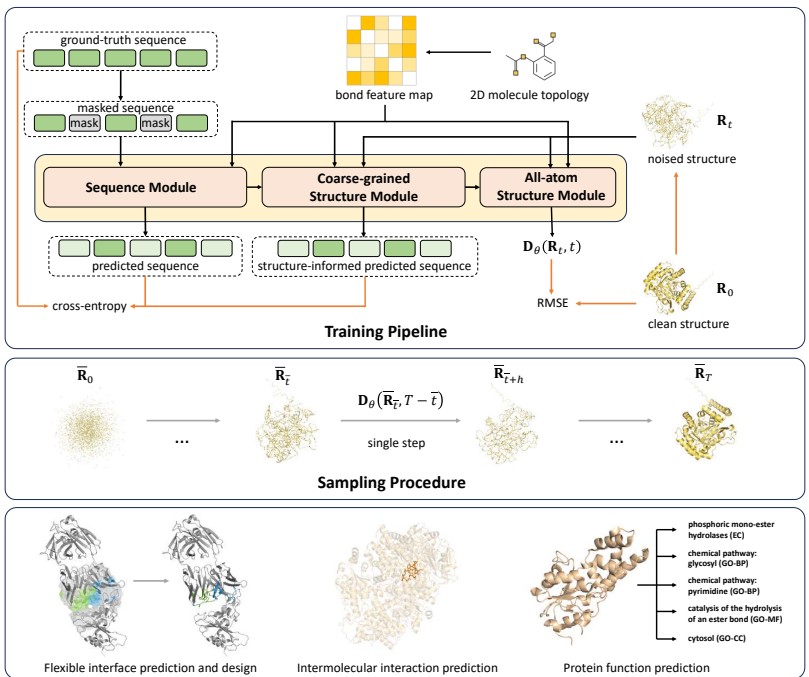

Figure 1: FlexRibbon framework. The model architecture consists of three modules: (i) a sequence module that encodes masked protein sequences and ligand topologies, (ii) a coarse-grained structure module that encodes residue-level structural information, and (iii) an all-atom structure module that refines these representations into chemically consistent coordinates. The framework combines diffusion-based denoising with sequence recovery, enabling joint alignment of sequence, residue, and atomic representations for complex modeling. Downstream tasks, including antibody/nanobody design, intermolecular interactions modeling and protein function prediction, are supported.

grained modeling captures global protein-ligand organization, while the all-atom stage ensures fine-grained structural accuracy. Table 4 shows the architectural hyperparameters.

**Sequence Module.** The sequence module jointly embeds protein residues and small-molecule atoms into a unified representation space. For protein residues, we apply a standard Transformer encoder (Appendix B) with rotary position embeddings (Su et al., 2023), focusing purely on sequence-derived semantics. To incorporate covalent-bond information of small molecules, we apply a small MLP to the atom-type embeddings to produce a 2D bond feature matrix. This learned projection enables the model to recover the covalent bonding pattern from atom identities through a single MLP layer, rather than relying on handcrafted bond encodings. The combined representations define a residue-atom graph, which is further refined by a pair-feature update module that models residue-residue and residue-atom interactions.

**Coarse-grained Structure Module.** The coarse-grained structure module employs a Diffusion Transformer (DiT) (Peebles & Xie, 2023) to denoise coordinates at residue level for proteins and atom level for small molecules. Each residue is represented as a coarse structural anchor, while each ligand atom is represented by a position embedding derived from its noised coordinates. The module conditions on embeddings from the sequence module to guide denoising.

**All-atom Structure Module.** The all-atom structure module employs a DiT where each atom of proteins is represented explicitly. Noised 3D coordinates of all atoms are encoded into position features that serve as token inputs. The coarse-grained outputs are broadcast to all atoms of each residue, providing residue-level guidance as conditional input. To preserve chemical validity, learnable attention biases are added to atom pairs connected by covalent bonds, combining atom-type and bond-type embeddings as additive bias terms in the attention map. This refinement stage allows the model to reconcile global residue-level context with detailed atomic-level interactions, yielding chemically consistent and high-resolution structures.

### 3.3 Structure-Informed Masked Language Model (SIMLM)

Masked language modeling (MLM) (Kenton & Toutanova, 2019; Lin et al., 2023) has proven effective for predicting masked amino acids in protein sequences. In the spirit of unifying sequence and structure, the masked positions should be inferred from correlations within the surrounding sequence and reflect the structural context that these residues possess. To realize this principle, we extend MLM beyond sequence-only inputs by integrating diffusion-based noise into structural representations, yielding a structure-informed masked language model (SIMLM). This formulation couples sequence recovery with structural denoising, thereby reinforcing the mapping between amino acid identity and three-dimensional conformation.

Concretely, we integrate MLM and diffusion through three complementary training modes. **Mode 1 (Sequence-to-Structure):** standard diffusion-based structure reconstruction, where clean sequences condition the generation of noisy structures. **Mode 2 (Coupled Perturbation):** for 15% of residues selected at random, mask the amino acid type and add diffusion noise to their local structures, while leaving all other tokens and structures unperturbed. **Mode 3 (Sequence-Masked Global Perturbation):** randomly select 15% of residues for type masking, while applying diffusion noise to the structures of all residues.

Through these modes, the model alternates between one-way mapping, localized joint perturbation, and global perturbation, which together encourage robust learning of the bidirectional relationship between protein sequences and structures. These allow the model to capture not only sequence-level regularities but also the structural constraints and variability that underlie protein evolution and function. More details are provided in Appendix C.3.

### 3.4 Training and Sampling

**Loss function.** Our training objective integrates four complementary components to balance coordinate accuracy, sequence recovery, and structural plausibility. The overall loss is defined as

$$\mathcal{L} = \mathcal{L}_{\text{MSE}} + \mathcal{L}_{\text{MLM}} + \mathcal{L}_{\text{Dist}} + \mathcal{L}_{\text{smooth-lDDT}}.$$

Here, $\mathcal{L}_{\text{MSE}}$ denotes the diffusion pretraining loss function given in Eq. (2), $\mathcal{L}_{\text{MLM}}$ improves sequence-level representation through masked residue prediction, $\mathcal{L}_{\text{Dist}}$ regularizes predicted inter-residue distances to maintain realistic tertiary structure geometry, and $\mathcal{L}_{\text{smooth-lDDT}}$ aligns training with widely used structure quality metrics by emphasizing local geometric accuracy.

**Training.** Pretraining is organized into two progressive stages. Stage A optimizes all components except $\mathcal{L}_{\text{MLM}}$, training on proteins with up to 384 residues. Deferring MLM at this stage avoids the instability that arises when it is introduced too early, while the residue cap improves efficiency and helps the model prioritize learning core structural regularities. Stage B expands the input length to 768 residues and incorporates $\mathcal{L}_{\text{MLM}}$, enabling stable joint optimization of sequence and structure on larger scales. In both stages, we include a confidence-weighted diffusion loss that scales residue-level contributions by pLDDT-derived sigmoid weights, reducing noise from low-confidence regions while emphasizing reliable structural signals. Stage C further extends the maximum sequence length to 1024 residues and introduces training of the confidence head, allowing the model to learn calibrated residue-level uncertainty estimates alongside structure generation. By jointly optimizing structural, sequence, and confidence objectives at full sequence scale, Stage C consolidates earlier-stage representations and equips the model with robust long-range reasoning ability and reliable self-assessment. More details are provided in Appendix C.5.

**Sampling.** The sampling procedure is the simulation of the reverse process. By leveraging the relation of the denoising model to the score model in Eq. (1), we have:

$$\bar{\mathbf{R}}_0 \sim p_{\text{src}} = \mathcal{N}(\mathbf{0}, \sigma_T^2 \mathbf{I}),$$

$$\bar{\mathbf{R}}_{\bar{t}+h} = \bar{\mathbf{R}}_{\bar{t}} + \frac{\mathbf{D}_\theta(\bar{\mathbf{R}}_{\bar{t}}, t) - \bar{\mathbf{R}}_{\bar{t}}}{\sigma_{T-\bar{t}}}(\sigma_{T-\bar{t}} - \sigma_{T-\bar{t}-h}). \tag{3}$$

We follow similar modifications as used in AlphaFold 3 (Abramson et al., 2024; Karras et al., 2022), but forgo applying the random rotation at each sampling step as orientation alignment is not used in the loss function. Hence, the model learns the correct output orientation relative to the input. The detailed sampling algorithm is presented in Appendix C.7.

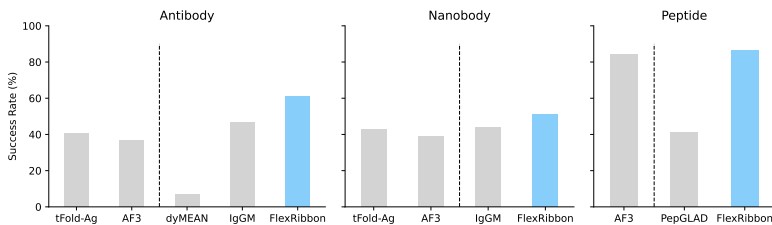

Figure 2: Success rates of structure prediction for antibodies, nanobodies, and peptides. tFold-Ag and AlphaFold 3 take MSA information as input. IgGM, dyMEAN, and FlexRibbon leverage antigen structural information. All methods, except AlphaFold 3, additionally incorporate epitope information. Results except FlexRibbon are taken from Wang et al. (2025a).

## 4 EXPERIMENTS

We use entries from the AlphaFold Protein Structure Database (AFDB, CC-BY 4.0 License) (Varadi et al., 2024) and the Protein Data Bank (PDB, CC0 1.0 License) (Berman et al., 2000) released on or before 2021-09-30 for pretraining (Appendix A). For downstream evaluation, we consider three major task families: (i) **Flexible interface prediction and design**, spanning five tasks involving antigen–antibody, antigen–nanobody, and protein–peptide complexes; (ii) **Intermolecular interaction prediction**, including three tasks centered on protein–ligand binding; and (iii) **Protein function prediction**, comprising four tasks focused on functional annotation. In addition, we employ a confidence head that predicts pLDDT and PAE; following AlphaFold 3, we compute pTM directly from the predicted PAE to provide a consistent estimate of global structural accuracy.

### 4.1 FLEXIBLE INTERFACE PREDICTION AND DESIGN

Biomolecules with flexible binding interfaces are difficult to model and design (Wu et al., 2025). Antibodies, nanobodies, and peptides are key examples, as their functions depend on flexible binding (Wu et al., 2023). This flexibility allows them to target diverse molecules, but also makes structure prediction challenging. To study this problem, we introduce tasks on flexible interface modeling, including antigen-antibody, antigen-nanobody, and protein-peptide complexes. Each task is defined as: given the sequence or structure of the components, predict the structure of the complex. To avoid overlap, protein chains in the test sets share at most 40% sequence identity with the training data. We focus on two tasks: interface structure prediction and interface design.

**Antibody and nanobody interface prediction** is a cornerstone of structural immunology, as accurate modeling underpins antibody discovery and therapeutic engineering. Nanobodies can be regarded as single-domain antibodies derived from VHH fragments (Harmsen & De Haard, 2007), allowing both classes to be modeled within a shared framework. For this task, we follow the evaluation procedure of IgGM (Wang et al., 2025a), measuring performance by the success rate (SR) based on the DockQ (Mirabello & Wallner, 2024) score, with a threshold of DockQ $\geq$ 0.23. Experiments are conducted on the SAb23H2 test set from IgGM, where we compare FlexRibbon against the structure prediction models AlphaFold 3 and tFold-Ag (Wu et al., 2024a), as well as the antibody design methods dyMEAN (Kong et al., 2023b) and IgGM. Following the IgGM protocol, we predict antigen-antibody (-nanobody) complex structures given the antigen sequence and antibody (nanobody) sequence. As shown in Fig. 2, FlexRibbon achieves success rates of 61.3% for antigen-antibody and 51.1% for antigen-nanobody complexes, yielding absolute improvements of 14.6% and 7.1% over IgGM, respectively. These results demonstrate that FlexRibbon effectively models antigen-antibody and antigen-nanobody interactions (Tables 7 and 8).

**Protein-peptide interface prediction** is another important scenario, as peptides often act as recognition motifs or regulators for diverse cellular processes. We use FoldBench (Xu et al., 2025) as the benchmark. We follow the FoldBench evaluation protocol, also reporting the success rate based on DockQ. FlexRibbon is compared with the structure prediction model AlphaFold 3 and the peptide design method PepGLAD (Kong et al., 2025). As shown in Fig. 2, FlexRibbon achieves an SR of 91.4%, exceeding AlphaFold 3 and PepGLAD by 7.0% and 10.2%, respectively. These findings suggest that FlexRibbon generalizes well to flexible peptide-protein binding scenarios (Table 9).

Table 1: Metrics for antibody and nanobody design. Both IgGM and FlexRibbon leverage antigen structure and epitope information. All baseline results are taken from Wang et al. (2025a).

| Method | Antibody | | | Nanobody | | |
|---|---|---|---|---|---|---|
| | H3-AAR | DockQ | SR | H3-AAR | DockQ | SR |
| dyMean | 0.294 | 0.079 | 0.049 | - | - | - |
| DiffAb (AF3) | 0.226 | 0.208 | 0.368 | 0.156 | 0.211 | 0.346 |
| IgGM | 0.360 | 0.246 | 0.433 | 0.183 | **0.267** | 0.415 |
| FlexRibbon | **0.414** | **0.273** | **0.460** | **0.218** | 0.244 | **0.437** |

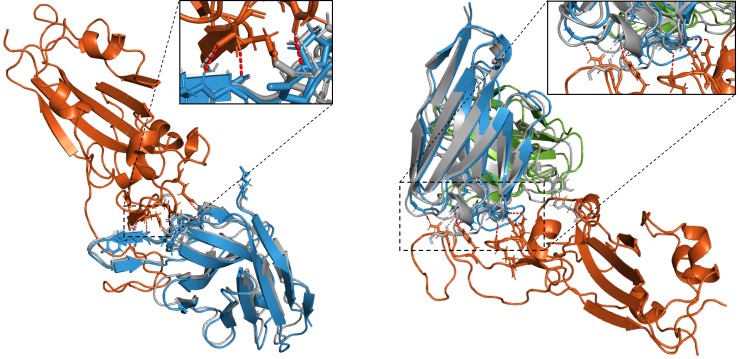

Figure 3: Predicted structures for (left) SARS-CoV-2 RBD with the Re30H02 nanobody (or our design) and (right) SARS-CoV-2 Omicron BA.4 RBD along with its antibody (or our design). Native structures are shown in grey, while those for sequences generated by FlexRibbon are in colour. The insets show close-up views of the interaction region, with dashed lines indicating presumed hydrogen bonds. For SARS-CoV-2 RBD (left), the native and designed sequences are structurally similar, with our design manifesting one additional hydrogen bond, whereas for SARS-CoV-2 Omicron BA.4 RBD (right), the predicted structures differ significantly in the interaction region, with our design yielding a greater number of hydrogen bonds. These showcases demonstrate the ability of FlexRibbon to generate sequences that are structurally sound.

**Antibody and nanobody design** is a key challenge in developing novel binders for therapeutic and diagnostic applications. In this task, the input is the antigen sequence and structure, and the objective is to design sequences for all antibody/nanobody CDR regions while jointly generating the full complex structure. We evaluate FlexRibbon on the SAb23H2 test set from IgGM, following the same protocol. Performance is measured by amino acid recovery (AAR), DockQ, and success rate (SR, defined as the proportion of samples which have DockQ $\geq$ 0.23) relative to wild-type complexes. Baselines include dyMEAN, DiffAb (Luo et al., 2022), and IgGM. As shown in Table 1, FlexRibbon achieves 41.4% AAR and 46.0% SR for antibody design, surpassing IgGM. For nanobody design, FlexRibbon also slightly outperforms IgGM, with higher AAR (21.8%) and SR (43.7%). In addition, FlexRibbon supports user-specified CDR lengths, enabling flexible design. Figure 3 illustrates some representative designs: our framework generates up to six CDRs simultaneously, with AAR reported specifically for the highly flexible CDR-H3 region, which is also the most challenging to design. Detailed results are provided in Appendix D. Together, these results show that FlexRibbon can generate realistic CDR sequences while maintaining structural fidelity to wild-type complexes.

## 4.2 INTERMOLECULAR INTERACTION PREDICTION

Protein-ligand interactions are fundamental in understanding protein conformational changes, binding affinities, and diverse biological functions. The accurate prediction of these interactions is thus crucial for elucidating molecular mechanisms and accelerating drug discovery. We evaluate FlexRibbon on three downstream tasks: protein-ligand docking prediction, ligand-induced conformational changes and protein-ligand binding affinity, which sharing a common foundation in modeling protein-ligand complexes, while emphasizing different aspects of interaction.

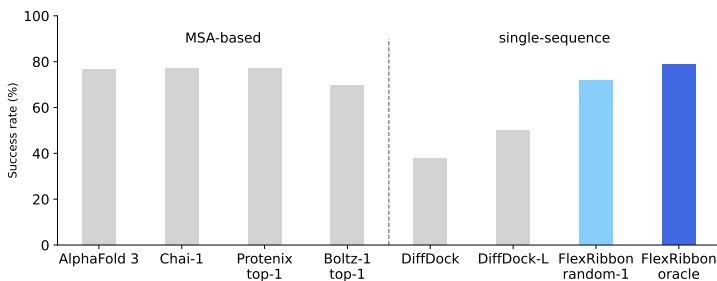

Figure 4: Evaluation of protein-ligand docking on the PoseBusters V1 benchmark. The methods are separated into (left) MSA-based and (right) single-sequence groups. The success rate is defined as the percentage of predictions with pocket-aligned ligand RMSD <2 Å. Apart from DiffDock and DiffDock-L, which predict the ligand pose with the protein structure given, all other methods jointly generate the structure of the protein-ligand complex. Results for AlphaFold 3 are taken from Jumper et al. (2021), Chai-1 from Chai Discovery (2024), DiffDock from Corso et al. (2023), and DiffDock-L from Corso et al. (2024). Results for Protenix, Boltz-1 and FlexRibbon were generated locally using a single seed with five generated samples. Other methods report top-1 ranked predictions, while our model does not use a confidence head; thus, we report random single-sample performance, with "oracle" denoting the best prediction among five samples selected against the ground truth.

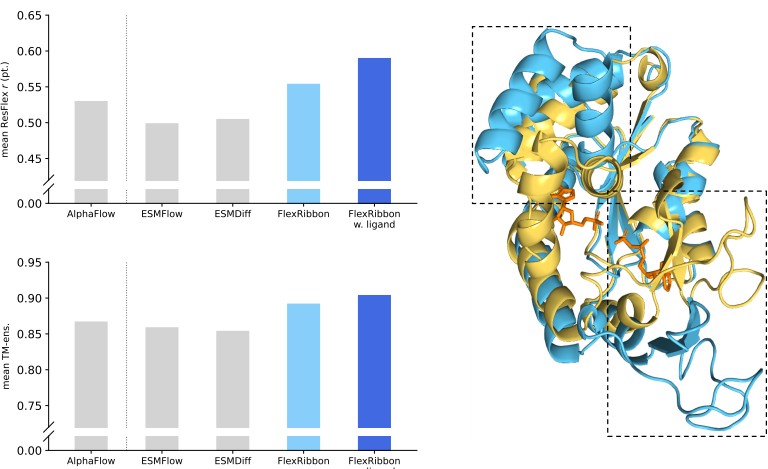

Figure 5: Evaluation of ligand-induced conformational change prediction. The left panels show the per-target mean correlations (top) and mean ensemble TM-scores (bottom). In the right panel, an overlay of the predicted apo (blue, PDB 4AKE) and holo (yellow, PDB 2ECK) structures of adenylate kinase is presented, illustrating the conformational changes (highlighted by the dashed boxed regions) induced by the presence of AMP and ADP molecules (orange). FlexRibbon predicts both states, with TM-score (Zhang & Skolnick, 2004) of 0.985 and 0.984, respectively.

**Protein-ligand docking prediction** is a key task for modeling intermolecular interactions with broad implications for life sciences and drug discovery. We follow the AlphaFold 3 protocol on the PoseBusters V1 benchmark (Buttenschoen et al., 2024) of 428 protein-ligand complexes, comparing against MSA-based models (AlphaFold3, Chai-1 (Chai Discovery, 2024), Protenix (ByteDance AML AI4Science Team et al., 2025), Boltz-1 (Wohlwend et al., 2025)) and single-sequence-based models (DiffDock (Corso et al., 2023), DiffDock-L (Corso et al., 2024)). Unlike previous single-sequence-based methods, which assume a fixed protein structure and only generate ligand poses, FlexRibbon jointly predicts protein-ligand structures directly from sequence like AlphaFold 3. As shown in Fig. 4, FlexRibbon achieves 71.82% in the random-1 regime and 78.70% under oracle selection, surpassing all single-sequence baselines by substantial margin and reaching parity with MSA-based approaches (Table 12).

Table 2: Evaluation of protein-ligand binding affinity prediction on CASF-2016.

| Method | RMSE ($\downarrow$) | R ($\uparrow$) |
|---|---|---|
| SIGN | 1.316 | 0.797 |
| GIANT | 1.269 | 0.814 |
| SPIN | 1.258 | 0.826 |
| FlexRibbon | **1.150** | **0.848** |

Table 3: $F_1$ for the Enzyme Commission (EC) and Gene Ontology (GO) tasks. The GO task is comprised of biological process (BP), molecular function (MF), and cellular component (CC).

| Method | EC | GO-BP | GO-MF | GO-CC |
|---|---|---|---|---|
| ESM-2-3B | 0.863 | 0.476 | 0.659 | 0.497 |
| SFM-Protein-3B | 0.869 | 0.495 | 0.673 | 0.510 |
| ESM-GearNet | 0.890 | 0.488 | 0.681 | 0.464 |
| FlexRibbon | **0.891** | **0.539** | **0.694** | **0.560** |

**Ligand-induced conformational change prediction** is key to understanding how proteins adapt upon ligand binding. We follow the ESMDiff protocol (Lu et al., 2025) and use the Apo/Holo dataset (Saldaño et al., 2022) to evaluate FlexRibbon with ensemble TM-score (TM-ens) and residue flexibility correlations (ResFlex $r$) at both global and per-target levels. Baselines include AlphaFlow (MSA-based), ESMFlow (Jing et al., 2024) (sequence-based), and ESMDiff (structure–language). Unlike these methods, FlexRibbon can model protein-ligand complexes in two modes: (i) protein-only (5 samples) and (ii) mixed (3 apo + 2 holo samples) with ligand guidance. Using a zero-shot pretrained checkpoint, FlexRibbon achieves a TM-ens score of 0.889 (improving upon ESMDiff by 0.038) and stronger flexibility correlations. Adding ligands further improves TM-ens by 0.012, showing the importance of ligand context. Figure 5 and Table 13 summarize the results.

**Protein-ligand binding affinity prediction** is a cornerstone of drug discovery, enabling efficient prioritization of candidate compounds for therapeutic targets. Traditional high-throughput screening is costly and limited in scope, motivating computational approaches that estimate binding affinities directly from protein-ligand structures. We evaluate FlexRibbon on the CASF-2016 benchmark (Su et al., 2018), using the standard metrics of root mean square error (RMSE) and Pearson's correlation coefficient ($R$). Comparisons are made against state-of-the-art baselines SIGN (Li et al., 2021), GLANT (Li et al., 2023), and SPIN (Choi et al., 2024). As shown in Table 2, FlexRibbon achieves the best performance on both criteria. These results highlight the value of pretrained embeddings derived from joint protein-ligand structures as a strong basis for accurate affinity prediction.

### 4.3 PROTEIN FUNCTION PREDICTION

Protein function prediction is central to characterizing novel proteins, understanding disease, and guiding therapeutic discovery. We evaluate this by finetuning FlexRibbon on Gene Ontology (GO) (Ashburner et al., 2000) and Enzyme Commission (EC) (Bairoch, 2000) annotation tasks using the DeepFRI (Gligorijević et al., 2021) setup. Baselines include sequence-only models (ESM-2-3B, SFM-Protein-3B (He et al., 2024)) and a sequence–structure hybrid (ESM-GearNet (Zhang et al., 2023)). Unlike these methods, FlexRibbon jointly predicts structures and embeddings without external structural input. Performance is measured by maximum $F_1$ score.

**EC number prediction** provides a controlled benchmark for catalytic function annotation, formulated as a binary classification task. As shown in Table 3, FlexRibbon achieves an $F_1$ score of 0.891, slightly exceeding ESM-GearNet and clearly outperforming sequence-only baselines (ESM-2 and SFM-Protein). These results highlight the importance of structural information in accurately capturing enzymatic function.

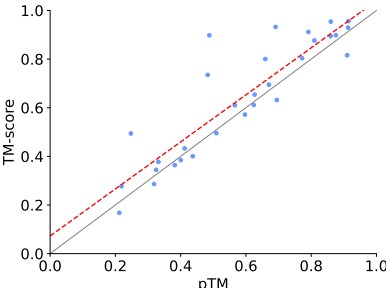 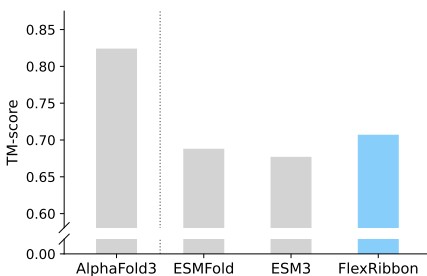

Figure 6: Confidence estimation and structure quality on the CASP15 benchmark. **Left:** Correlation between observed TM-score and predicted pTM, with a Pearson correlation coefficient of 0.89. For each target, 25 candidate structures are generated, and the highest-confidence sample is used to compute the chain-level TM-score. **Right:** TM-scores of the highest-confidence predicted structures compared with their corresponding native structures.

**GO term prediction** evaluates protein function across biological processes (BP), molecular functions (MF), and cellular components (CC), each framed as an independent multi-label classification task, consistent with the EC setup. As shown in Table 3, FlexRibbon achieves F1 scores of 0.539 (BP), 0.694 (MF), and 0.560 (CC), outperforming ESM-GearNet by 0.051, 0.013, and 0.096, respectively. These gains indicate that FlexRibbon provides more informative representations for finetuning, enabling more accurate functional annotation across ontologies.

### 4.4 CONFIDENCE HEAD

Confidence estimation is essential for evaluating the reliability of predicted protein structures. Following AlphaFold 3, our confidence head predicts the pairwise alignment error (PAE), and we compute the pTM score from these PAE values using the same formulation. This allows the model to produce a global fold-accuracy estimate (pTM) that reflects its learned uncertainty distribution. As shown in Fig. 6, the predicted pTM correlates strongly with the true TM-score on the CASP15 (Kryshtafovych et al., 2023) benchmark, achieving a Pearson correlation of $R = 0.89$.

For protein monomer structure prediction, we generate 25 samples per target and select the structure with the highest confidence score for evaluation. As shown in Fig. 6, FlexRibbon achieves a TM-score of 0.703 on CASP15 benchmark, surpassing the single-sequence baselines ESMFold and ESM3 by 0.019 and 0.030 TM-score, respectively. These results demonstrate the reliability of our confidence estimation and the strong effectiveness of the overall structure prediction. Additional details are provided in Appendix D.9.

## 5 CONCLUSION

We introduce FlexRibbon, a pretrained protein foundation model that integrates both sequence and structural information into a unified framework. Unlike prior approaches that impose a one-way sequence-to-structure mapping, FlexRibbon implements joint training via masked language modeling and diffusion-based denoising, enabling bidirectional sequence-structure representations that support both prediction and design. Extensive evaluations across a diverse set of downstream tasks spanning antibody/nanobody and peptide interface modeling, ligand-induced conformational change, protein-ligand binding affinity, and functional annotation demonstrate strong and consistent gains, with especially notable improvements on flexible and mutation-rich interfaces where existing methods struggle. These results highlight the effectiveness of joint sequence-structure pretraining and show that its benefits extend broadly beyond protein folding, establishing FlexRibbon as a general-purpose foundation model for protein science.

### ACKNOWLEDGMENTS

This work is supported by Zhongguancun Academy (Grant No. C20250514).

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

## A  DATA

**The pretraining dataset** is constructed from two primary sources: **AFDB** and **PDB**.

The AlphaFold Protein Structure Database (AFDB), released by Google DeepMind and EMBL-EBI, contains over 200 million predicted structures spanning nearly the entire **UniProt_2021_04** release. To reduce redundancy, we cluster sequences at 90% identity and retain only entries with a global pLDDT score greater than 50, discarding low-confidence structures. This yields approximately 78 million AFDB samples.

For experimentally resolved structures in Protein Data Bank (PDB), we use **PDB_20210930**, adopting the same cutoff date as AlphaFold 3. Following their filtering protocol, we exclude structures with more than 300 chains, resolution worse than 9 Å,or fewer than 4 residues. After filtering, we obtain roughly 181 thousand PDB samples.

In total, our pretraining corpus comprises more than 78 million protein structures.

The datasets used for each downstream task are detailed in Appendix D.

## B  ARCHITECTURE

Figure 7 and Table 4 summarize the model. FlexRibbon adopts a three-stage design:

**Sequence module.** Jointly embeds protein residues and small-molecule atoms into a unified space using a Transformer encoder. Protein residues receive RoPE for order sensitivity, while atom embeddings remain permutation-invariant. Learnable attention biases from atom categories and bond types provide chemical context, and a pair-feature update module refines residue–atom interactions. In addition, we introduce a 2D covalent-bond feature by embedding the ordered atom-type pairs associated with each potential bond. This yields a pairwise tensor of shape $(B, L, L, H_{\mathrm{bf}})$ that encodes bond-aware chemical relations across all atom pairs. A lightweight MLP projects this representation to a single scalar per pair, which is added to the attention logits as an additional structural prior. This learned 2D bond feature enables the model to capture covalent connectivity in a differentiable manner, complementing both sequence embeddings and atom-type cues.

**Coarse-grained structure module.** Applies a Diffusion Transformer (DiT) to denoise residue and atom coordinates at reduced resolution. Conditioning on sequence embeddings integrates sequence

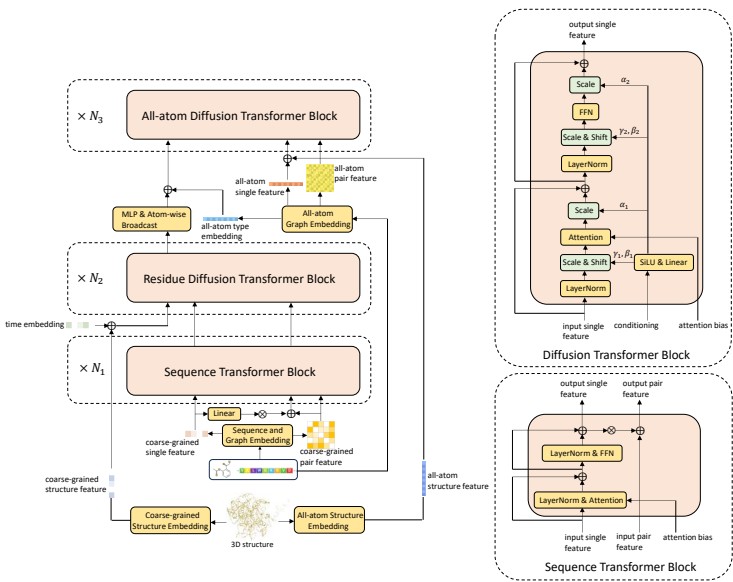

Figure 7: Detailed Model Architecture of FlexRibbon.

semantics with structural noise while keeping attention maps small enough to allocate parameters to structural reasoning.

**All-atom structure module.** Upsamples to full atomic resolution with a second DiT. Coarse outputs are broadcast to all atoms in each residue as conditional input. Atom- and bond-type biases are added to attention maps to preserve chemical validity, reconciling residue-level context with detailed atomic geometry.

This design captures the global geometry of diverse biomolecular complexes and refines them into chemically consistent, high-resolution structures. Key hyperparameters are listed in Table 4.

Table 4: Key architectural hyperparameters of FlexRibbon.

| Component | Hyperparameter | Value |
|---|---|---|
| Sequence Module | Number of layers | 32 |
| | Hidden size | 2,048 |
| | FFN dimension | 8,192 |
| | Attention heads | 32 |
| Coarse-grained structure module | Number of layers | 16 |
| | Hidden dimension | 2,048 |
| | FFN dimension | 8,192 |
| | Attention heads | 32 |
| All-atom structure module | Number of layers | 8 |
| | Embedding dimension | 256 |
| | FFN dimension | 256 |
| | Attention heads | 4 |

## C    TRAINING AND INFERENCE

### C.1    CONFIDENCE-WEIGHTED DIFFUSION LOSS.

To incorporate structural reliability, we scale the diffusion MSE loss using residue-level pLDDT scores with a sigmoid weighting function. Residues in low-confidence regions (pLDDT around 60 or below) receive weights near zero, whereas residues in high-confidence regions (pLDDT around 80 or above) receive weights near one. The transition between these regimes is controlled by a smooth sigmoid:

$$w = \sigma\left(\beta \cdot \frac{\text{pLDDT} - 70}{10}\right),$$

where $\sigma(\cdot)$ is the logistic sigmoid and $\beta$ controls the steepness of the curve. In practice, we set $\beta = 5$ such that weights are near zero at pLDDT $= 60$ and near one at pLDDT $= 80$. This formulation avoids hard thresholds while ensuring that uncertain structural regions contribute less to the optimization, and high-confidence regions dominate the learning signal. Also, this design keeps signal from disordered regions without letting their noisy coordinates dominate the loss, a balance that aligns with the qualitative behavior shown in Figure 9 of Section D.11 using the disordered example reproduced from the AlphaFold3 paper.

### C.2    INTER-RESIDUE DISTANCE LOSS.

We apply $\mathcal{L}_{\text{Dist}}$ on top of the sequence module to regularize predicted inter-residue distances. Specifically, the sequence encoder outputs residue-level embeddings, which are combined through an outer product to form pairwise features. A lightweight MLP head then predicts the $C_\alpha$-$C_\alpha$ distance for each residue pair. The loss penalizes deviations between predicted and ground-truth distances, encouraging the encoder to capture geometric constraints directly at the sequence-pair feature level. This design provides the model with explicit supervision on tertiary structure geometry while avoiding direct reliance on coordinate-level regression.

### C.3 Structure-informed Masked Language Modeling (SIMLM) loss.

We design a structure-informed masked language modeling loss to align sequence and structure representations. Only protein sequences (FASTA) are masked, following the BERT-style policy: $15\%$ of residues are selected for corruption, with $80\%$ replaced by `[MASK]`, $10\%$ replaced by a random amino acid, and $10\%$ left unchanged. For each masked residue $i$ with ground-truth identity $y_i$, we compute hidden features from both the sequence encoder $f_i^{\text{seq}}$ and the coarse-grained structure encoder $f_i^{\text{struct}}$ (e.g., based on $C_\alpha$ geometry). Two independent prediction heads are applied: one maps $f_i^{\text{seq}}$ to a distribution $p_\theta^{(\text{seq})}(y_i)$ and the other maps $f_i^{\text{struct}}$ to $p_\theta^{(\text{struct})}(y_i)$. The loss averages the negative log-likelihoods from both heads:

$$\mathcal{L}_{\text{S-MLM}} = -\frac{1}{|\mathcal{M}|} \sum_{i \in \mathcal{M}} \left[ \log p_\theta^{(\text{seq})}(y_i \mid f_i^{\text{seq}}) + \log p_\theta^{(\text{struct})}(y_i \mid f_i^{\text{struct}}) \right],$$

where $\mathcal{M}$ is the set of masked positions. This formulation encourages both the sequence and structure pathways to retain predictive signal for residue identity, thereby improving cross-modal consistency. In practice, we interleave the three perturbation modes (Mode 1, Mode 2, and Mode 3) during training with a ratio of $6:2:2$, balancing standard sequence-to-structure generation with increasingly challenging coupled and global perturbations.

### C.4 Smooth-lDDT loss.

Following AlphaFold2 (Jumper et al., 2021), we compute the smooth local distance difference test (lDDT) loss to assess local structural accuracy. The smooth lDDT metric measures the agreement of predicted inter-residue distances with the ground truth in a differentiable manner. Specifically, for each residue $i$, we evaluate all neighboring residues $j$ within a cutoff radius (typically 15 Å). For each pair $(i,j)$, the absolute deviation of the predicted $C_\alpha$-$C_\alpha$ distance from the reference is mapped to a soft score using a piecewise linear function with thresholds at 0.5, 1, 2, and 4 Å. The residue-wise scores are averaged across neighbors and then across residues to produce the overall smooth lDDT. In training, we only use $C_\alpha$ atoms to compute this loss, consistent with AlphaFold2. The resulting value serves both as a differentiable accuracy proxy and as a regularizer encouraging the model to capture local geometric consistency.

### C.5 Training Recipe.

We adopt a three-stage pretraining strategy (see Table 5). Stage A focuses on diffusion-based denoising with proteins of up to 384 residues. Training uses the Adam optimizer (Kingma & Ba, 2015) in bfloat16 mixed precision with a batch size of 4,096 on 128 A100 GPUs for 200k steps and a learning rate of $1 \times 10^{-4}$. This stage builds the core ability to reconstruct clean structures from noisy inputs while incorporating structural regularization via distance and smooth-lDDT losses. Stage B extends the maximum protein length to 768 residues and adds the masked language modeling (MLM) objective. Training uses a batch size of 2,048 on the same hardware for 100k steps with a learning rate of $6 \times 10^{-5}$. This stage enables the model to handle larger proteins and integrate sequence-level supervision, while continuing to optimize diffusion, distance, and smooth-lDDT objectives.

Stage C further increases the maximum sequence length to 1,024 residues and introduces supervised training of the confidence head. Following AlphaFold 3 (Abramson et al., 2024), we generate confidence labels by running 20-step mini rollout sampling to obtain predicted structures and computing pLDDT and PAE targets from these rollouts. In our architecture, the confidence head is implemented as a lightweight MLP attached after the coarse-grained structure module, taking its structure-informed residue embeddings as input to predict PAE and pLDDT. Unlike AlphaFold 3, however, we do not stop gradients from the confidence losses; instead, we allow them to update sequence module and coarse-grained structure module's parameters. Our confidence head is a lightweight MLP rather than a specialized architecture. Stage C is trained for 20k steps with a batch size of 2,048 and a learning rate of $2 \times 10^{-5}$, consolidating joint optimization of structure, sequence, and confidence estimation at full sequence scale.

### C.6 Diffusion training details

We provide here the complete derivations and formulation details omitted from the main text.

Table 5: Three-stage pretraining configuration.

|  | Stage A | Stage B | Stage C |
|---|---|---|---|
| Max residues | 384 | 768 | 1024 |
| MSE Loss | True | True | True |
| Dist Loss | True | True | True |
| Smooth lDDT Loss | True | True | True |
| MLM Loss | False | True | True |
| plDDT Loss | False | False | True |
| PAE Loss | False | False | True |
| Batch size | 4,096 | 2,048 | 2,048 |
| Steps | 200k | 100k | 20k |
| Learning rate | $1 \times 10^{-4}$ | $6 \times 10^{-5}$ | $2 \times 10^{-5}$ |

**Forward process.** Diffusion-based generative modeling aims to approximate a target distribution $p(\mathbf{R})$ by connecting it to a tractable source distribution $p_{\text{src}}$. We represent a protein structure $\mathbf{R} \in \mathbb{R}^{3N}$ by the 3D coordinates of all heavy atoms. The forward noising process is defined as

$$\mathbf{R}_t = \mathbf{R}_0 + \sigma_t \boldsymbol{\epsilon}_t, \quad \boldsymbol{\epsilon}_t \sim \mathcal{N}(\mathbf{0}, \mathbf{I}), \quad t \in [0, T],$$

with $\sigma_t$ monotone increasing in $t$. For sufficiently large $\sigma_T$, $\mathbf{R}_T$ approximates a Gaussian distribution $p_{\text{src}} = \mathcal{N}(\mathbf{0}, \sigma_T^2 \mathbf{I})$.

This corresponds to the SDE

$$d\mathbf{R}_t = \sqrt{(\sigma_t^2)'} \, d\mathbf{w}_t,$$

where $\mathbf{w}_t$ is a Wiener process on $\mathbb{R}^{3N}$.

**Reverse process.** By stochastic process theory (Anderson, 1982), one can recover $p(\mathbf{R})$ by simulating the reverse diffusion process. A deterministic equivalent is given by the probability-flow ODE (Song et al., 2021):

$$d\bar{\mathbf{R}}_{\bar{t}} = \tfrac{1}{2}(\sigma_t^2)'|_{t=T-\bar{t}} \nabla \log p_{T-\bar{t}}(\bar{\mathbf{R}}_{\bar{t}}) \, d\bar{t},$$

where $\bar{t}$ denotes reversed time and $\bar{\mathbf{R}}_{\bar{t}}$ denotes the reversed sample trajectory.

**Score estimation.** The only unknown term is $\nabla \log p_t$, which we approximate with $\mathbf{s}_\theta(\mathbf{R}, t)$. Minimizing the score-matching objective

$$\mathbb{E}_{p_t(\mathbf{R}_t)} \|\mathbf{s}_\theta(\mathbf{R}_t, t) - \nabla \log p_t(\mathbf{R}_t)\|^2$$

is equivalent to a denoising objective with conditional distribution $p(\mathbf{R}_t \mid \mathbf{R}_0) = \mathcal{N}(\mathbf{R}_t \mid \mathbf{R}_0, \sigma_t^2 \mathbf{I})$:

$$\min_\theta \mathbb{E}_{p(t)} w_t \, \mathbb{E}_{p(\mathbf{R}_0)} \mathbb{E}_{p(\mathbf{R}_t \mid \mathbf{R}_0)} \|\mathbf{D}_\theta(\mathbf{R}_t, t) - \mathbf{R}_0\|^2,$$

where we use the parameterization

$$\mathbf{s}_\theta(\mathbf{R}, t) := \frac{\mathbf{D}_\theta(\mathbf{R}, t) - \mathbf{R}}{\sigma_t^2}.$$

Intuitively, the network predicts the clean structure $\mathbf{R}_0$ from its noisy version $\mathbf{R}_t$, hence the name "denoising model."[*]

**Rigid-body invariances.** Protein structures are equivalent up to rigid-body transformations. Translations are removed by centering at the center of mass. Rotational invariance is harder: while $\mathrm{SO}(3)$-equivariant networks (Köhler et al., 2020) can guarantee invariance, they often require heavy operations and introduce reflection symmetry. We instead use a standard architecture and provide rotational invariance information via random $\mathrm{SO}(3)$ data augmentation. Some works apply explicit rotational alignment in the loss (Xu et al., 2022; Abramson et al., 2024), but such alignment lacks a consistent orientation correspondence and complicates sampling (Wohlwend et al., 2025). In our experiments, the plain denoising objective already yielded stable and effective training, so we removed the alignment operation in the loss.

---

[*]Recovering the exact $\mathbf{R}_0$ is impossible due to information loss; the model in fact predicts $\mathbb{E}[\mathbf{R}_0 \mid \mathbf{R}_t]$.

---

**Algorithm 1** Sampling procedure.

---

**Require:** A trained diffusion model in denoising form $\mathbf{D}_\theta(\mathbf{R}, t)$ under the noise schedule choice $\sigma_t = t$, sampling time schedule $0 = \bar{t}_0 < \bar{t}_1 < \cdots < \bar{t}_N = T$, recursion ratio $\gamma_{\text{recur}}$, recursion threshold $\gamma_{\min}$, noise scale $\lambda$.
1: Initialize $\bar{\mathbf{R}}_0 \sim p_{\text{src}} := \mathcal{N}(\mathbf{0}, T^2\mathbf{I})$;
2: **for** $i = 0$ to $N - 1$ **do**
3:     Center $\bar{\mathbf{R}}_i$ to its center of mass;
4:     $\gamma = \gamma_{\text{recur}}$ if $T - \bar{t}_i > \gamma_{\min}$ else 0;
5:     $\hat{\bar{t}}_i = (1 + \gamma)\bar{t}_i - \gamma T$;
6:     $\bar{\mathbf{R}}_{\hat{\bar{t}}_i} = \bar{\mathbf{R}}_{\bar{t}_i} + \lambda\sqrt{(T - \hat{\bar{t}}_i)^2 - (T - \bar{t}_i)^2}\,\boldsymbol{\epsilon}$, where $\boldsymbol{\epsilon} \sim \mathcal{N}(\mathbf{0}, \mathbf{I})$;
7:     $\bar{\mathbf{R}}_{\bar{t}_{i+1}} = \bar{\mathbf{R}}_{\hat{\bar{t}}_i} + \frac{\mathbf{D}_\theta(\bar{\mathbf{R}}_{\hat{\bar{t}}_i}, T - \hat{\bar{t}}_i) - \bar{\mathbf{R}}_{\hat{\bar{t}}_i}}{T - \hat{\bar{t}}_i}(\hat{\bar{t}}_i - \bar{t}_{i+1})$;
8: **end for**
9: **return** $\bar{\mathbf{R}}_{\bar{t}_N}$.

---

## C.7 SAMPLING PROCEDURE

From the reverse sampling formulation in Eq. (3), what essentially controls the progression of the diffusion process is the discretization of $\sigma_t$. A convenient choice is thus to let $\sigma_t = t$ (Karras et al., 2022). The sampling process is then specified by a discretization of the reverse time $0 = \bar{t}_0 < \bar{t}_1 < \cdots < \bar{t}_N = T$, where $N$ is the number of discretization steps. Following (Karras et al., 2022) (which is also adopted in Alphafold 3 (Abramson et al., 2024)), in each step, the update starts not directly from the current time step $\bar{t}_i$. Instead, the clock is first recurred back to $\hat{\bar{t}}_i := (1+\gamma)\bar{t}_i - \gamma T$ (which comes from increasing the forward time by $(1 + \gamma)$, *i.e.*, $T - \hat{\bar{t}}_i = (1 + \gamma)(T - \bar{t}_i)$) with a more noisy state, which can be implemented by simulating the forward process from $T - \bar{t}_i$ to $T - \hat{\bar{t}}_i$ as $\bar{\mathbf{R}}_{\hat{\bar{t}}_i} = \bar{\mathbf{R}}_{\bar{t}_i} + \sqrt{(T - \hat{\bar{t}}_i)^2 - (T - \bar{t}_i)^2}\,\boldsymbol{\epsilon}$, where $\boldsymbol{\epsilon} \sim \mathcal{N}(\mathbf{0}, \mathbf{I})$. The simulation then proceeds by an update from $\hat{\bar{t}}_i$ to $\bar{t}_{i+1}$ following Eq. (3). In contrast to the sampling process by Alphafold 3, we do not need a random rotation in each step as we do not use rotational alignment in training. The complete procedure is presented in Alg. 1.

## C.8 INFERENCE TIME AND MEMORY

To assess the practicality of FlexRibbon for real-world downstream applications, we measure its inference time and memory consumption. We select four heteromers and four homomers from the PDB, with sequence lengths of approximately 256, 512, 1024, and 2048 residues as test cases. Note that the comparison focuses solely on the forward pass of the network and does not include data preprocessing time. All experiments are conducted on a single NVIDIA A100 (80 GB) GPU.

Table 6: Inference time and memory profile using single A100 80 GB.

| PDB | Type | Residue number | Inference time (sec.) | GPU Memory (GiB) |
|---|---|---|---|---|
| 1AB9 | Heteromer | 246 | 11.3 | 6.7 |
| 1KG0 | Heteromer | 517 | 16.7 | 8.0 |
| 8ODX | Heteromer | 1,024 | 43.3 | 11.0 |
| 8TQO | Heteromer | 2,064 | 139.8 | 27.0 |
| 4W80 | Homomer | 256 | 10.9 | 7.0 |
| 8VE4 | Homomer | 512 | 15.7 | 7.7 |
| 1VH1 | Homomer | 1,024 | 42.0 | 11.0 |
| 6EBM | Homomer | 2,052 | 139.0 | 27.2 |

# D    EXPERIMENTS DETAILS

Downstream tasks are application-specific benchmarks designed to evaluate how effectively a pre-trained foundation model can be adapted to solve targeted scientific problems. While pretraining provides the model with general sequence-structure representations, downstream tasks assess its transferability to practical domains such as protein design, intermolecular interaction prediction, and functional annotation. These tasks typically involve fine-tuning the model on smaller, curated datasets and comparing its performance against established baselines. By systematically evaluating across diverse downstream tasks, we demonstrate not only the generality of the pretrained model, but also its ability to capture biologically meaningful features that enable scientific discovery. Details of the fine-tuning procedure, dataset partitioning, and evaluation metrics, are provided below.

## D.1    ANTIBODY AND NANOBODY INTERFACE PREDICTION

**Datasets.** High-quality datasets are essential for evaluating antibody and nanobody interface modeling. We use SAbDab (Dunbar et al., 2013) as the training dataset and adopt the same training, validation, and test splits as in IgGM (Wang et al., 2025a) to ensure fair comparison with cutoff 2022-12-31. Moreover, we removed anti-ligand pattern from the dataset. In total, we constructed 10028 samples from 5146 unique PDB ids, in which 2023 samples are nanobody and make up 1108 unique PDB ids. for training and validation and evaluate performance on 60 antigen-antibody docking structures (SAb-23H2-Ab) and 27 antigen-nanobody docking structures (SAb-23H2-Nano).

**Finetuning and inference.** Accurate antibody modeling requires leveraging both sequence and structural information. We incorporate epitope annotations, which have been shown to be critical for reliable antibody prediction (Wang et al., 2025a), by labeling residues with at least one heavy atom within 10 Å of an antibody or nanobody chain. During fine-tuning, we adopt 4 complementary training modes to balance complex structure prediction and antibody design: (i) with 30% probability, the model receives full sequences and predicts the complex structure; (ii) with 40% probability, the model is provided with the antibody backbone sequence, antigen sequence, and antigen structure, and is tasked with designing antibody CDR sequences and structures; (iii) with 15% probability, the model receives antibody and antigen sequences along with the antibody structure and predicts the antigen structure; and (iv) with 15% probability, the model receives antibody and antigen sequences along with the antigen structure and predicts the antibody structure. Following the IgGM inference protocol, we generate 5 samples per test case. The input consists of antigen sequence, antigen structure, epitope annotations, and antibody sequence, and the model predicts the complex structure.

**Evaluation metrics.** Model performance is evaluated using DockQ, interface RMSD (iRMS), lig-and RMSD (LRMS), and success rate (SR, defined as DockQ $\geq 0.23$), which are widely used in the antibody modeling community (Mirabello & Wallner, 2024; Wu et al., 2024b; Wang et al., 2025a). DockQ, iRMS, and LRMS are averaged across all generated samples, while the success rate is computed as the fraction of all generated samples with DockQ $\geq 0.23$. As shown in Table 7 and Table 8, FlexRibbon consistently achieves the best performance across all metrics, substantially outperforming existing baselines.

Table 7: Metrics for prediction of antigen-antibody docking structure. tFold-Ag and AlphaFold 3 use MSA information as input. dyMEAN, IgGM, and FlexRibbon use antigen structure information. All methods except AlphaFold 3 use epitope information. AlphaFold 3, dyMEAN, tFold-Ag, and IgGM results are taken from Wang et al. (2025a). Methods marked with [†] use MSA as input.

| Method | DockQ↑ | iRMS↓ | LRMS↓ | SuccessRate↑ |
|---|---|---|---|---|
| tFold-Ag→HDock[†] | 0.022 | 16.652 | 48.157 | 0.0000 |
| tFold-Ag[†] | 0.252 | 6.796 | 21.035 | 0.4068 |
| AlphaFold 3[†] | 0.295 | 10.965 | 32.408 | 0.3684 |
| dyMEAN | 0.101 | 8.923 | 27.423 | 0.0667 |
| IgGM | 0.299 | 6.220 | 19.489 | 0.4667 |
| FlexRibbon | **0.384** | **5.704** | **14.533** | **0.6133** |

Table 8: Metrics for structure prediction for nanobody. tFold-Ag and AlphaFold 3 use MSA information as input. IgGM, and FlexRibbon use antigen structure information. All methods except AlphaFold 3 use epitope information. AlphaFold 3, tFold-Ag, and IgGM results are taken from Wang et al. (2025a). Methods marked with $^\dagger$ use MSA as input.

| Method | DockQ↑ | iRMS↓ | LRMS↓ | SuccessRate↑ |
|---|---|---|---|---|
| tFold-Ag$^\dagger$ | 0.288 | 6.349 | 15.081 | 0.4296 |
| AlphaFold 3$^\dagger$ | 0.287 | 11.219 | 32.676 | 0.3885 |
| IgGM | 0.291 | 7.988 | 22.017 | 0.4400 |
| FlexRibbon | **0.336** | **5.380** | **11.632** | **0.5111** |

## D.2 PROTEIN-PEPTIDE INTERFACE PREDICTION

**Datasets.** High-quality, non-redundant datasets are essential for training accurate protein-peptide interface models. We constructed the training dataset from PepGLAD (Kong et al., 2025) and applied a temporal filter to exclude entries released after September 30, 2021, ensuring that the pre-trained model had no prior exposure to test-like data. After filtering, the dataset contains 5,202 non-redundant protein-peptide complexes, reduced from the original 6,105 entries. For evaluation, we use FoldBench (Xu et al., 2025), which comprises 51 protein-peptide pairs, all sharing less than 40% sequence identity with the training and validation sets.

**Finetuning and inference.** Effective fine-tuning is crucial for adapting a foundation model to specific downstream tasks. We adopt the same hyperparameters as in the antibody and nanobody interface prediction task. During inference, we follow the AlphaFold 3 procedure, generating 5 samples per test instance. The input consists of the protein sequence, protein structure, epitope annotations, and the peptide sequence, and the model predicts the corresponding protein-peptide complex structure. Additional details are provided in Section D.1.

Table 9: Metrics for peptide structure prediction. AlphaFold 3 and Boltz-1 use MSA information as input, while PepGLAD and FlexRibbon leverage protein structure and epitope information. Results for Boltz-1 and AlphaFold 3 are taken from (Xu et al., 2025), where DockQ scores are not reported. PepGLAD results are obtained by running the method on this benchmark. Methods marked with $^\dagger$ use MSA as input.

| Method | DockQ↑ | iRMS↓ | LRMS↓ | SuccessRate↑ |
|---|---|---|---|---|
| Boltz-1$^\dagger$ | - | 2.88 | 8.82 | 0.8039 |
| AlphaFold 3$^\dagger$ | - | 2.81 | 6.56 | 0.8431 |
| PepGLAD | 0.413 | 3.18 | 6.72 | 0.8118 |
| FlexRibbon | **0.558** | **1.93** | **5.17** | **0.9137** |

**Evaluation metrics.** Standardized metrics are important for consistent assessment of interface prediction performance. We evaluate model performance using DockQ, interface RMSD (iRMS), ligand RMSD (LRMS), and success rate (SR, defined as the proportion of samples which have DockQ $\geq 0.23$). DockQ, iRMS, and LRMS are averaged across all generated samples, while the success rate is computed as the fraction of samples with DockQ $\geq 0.23$. When computing DockQ, the heavy and light chains of the antibody are merged into a single chain, with the antigen treated as a separate chain. As shown in Table 9, FlexRibbon outperforms all other methods across these metrics, demonstrating more accurate modeling of protein-peptide interfaces.

## D.3 ANTIBODY AND NANOBODY DESIGN

**Datasets.** High-quality and consistent datasets are essential for evaluating CDR design performance. We use the same training, validation, and test datasets as described in Section D.1 to ensure comparability and reproducibility.

**Training and inference.** Effective CDR design requires careful integration of sequence and structural information. During training, we follow the procedure in Section D.1. At inference, the CDR regions are masked to enable the model to design new sequences based on the antigen structure and epitope annotations. Users can also specify different CDR lengths to generate diverse designs. Following the IgGM evaluation protocol, we generate 5 samples per test case using the same CDR lengths as the wild-type sequences, jointly designing all six CDR regions for antibodies and all three CDR regions for nanobodies.

Table 10: Comparison of antibody modeling methods for antibody design, reporting CDR loop accuracy (AAR, RMSD) and docking performance. Higher values of AAR, DockQ, and SR indicate better performance, while lower values of RMSD, iRMS, and LRMS are preferable. DockQ scores are computed by comparing the designed structures against the corresponding wild-type complexes.

| Model | DiffAb (IgFold) | dyMEAN | IgGM | FlexRibbon |
|---|---|---|---|---|
| **AAR ↑** | | | | |
| L1 | 0.597 | 0.633 | **0.750** | 0.727 |
| L2 | 0.598 | 0.634 | 0.743 | **0.773** |
| L3 | 0.421 | 0.570 | 0.635 | **0.653** |
| H1 | 0.642 | 0.742 | 0.740 | **0.745** |
| H2 | 0.363 | 0.627 | 0.644 | **0.683** |
| H3 | 0.214 | 0.294 | 0.360 | **0.414** |
| **Docking with wild-type** | | | | |
| DockQ ↑ | 0.022 | 0.079 | 0.246 | **0.273** |
| iRMS ↓ | 17.034 | 9.698 | **6.579** | 6.961 |
| LRMS ↓ | 48.163 | 28.764 | 19.678 | **19.599** |
| Success Rate ↑ | 0.000 | 0.049 | 0.433 | **0.460** |

**Evaluation metrics.** To assess both sequence fidelity and structural accuracy in CDR design, we employ standard quantitative metrics. Sequence similarity to the wild type is measured using amino acid recovery (AAR) (Wang et al., 2025a), where higher values indicate closer agreement. For antibodies, AAR is computed independently for each of the six CDRs (three heavy-chain and three light-chain regions) and then averaged across samples; for nanobodies, AAR is computed over the three CDRs. Structural quality is evaluated using DockQ, interface RMSD (iRMS), ligand RMSD (LRMS), and success rate (SR; the percentage of designs with DockQ $\geq 0.23$). DockQ is computed by merging the antibody heavy and light chains into a single chain and treating the antigen as a separate chain; for nanobodies, DockQ is computed between the nanobody and the antigen. We compare against established baselines including DiffAb (Luo et al., 2022), dyMEAN(Kong et al., 2023b), and IgGM (Wang et al., 2025a) over the evaluation metrics. As shown in Tables 10 and 11, FlexRibbon achieves performance comparable to IgGM across all metrics, indicating that it generates realistic CDR sequences while preserving structural accuracy.

Table 11: Comparison of nanobody modeling methods for nanobody design, reporting CDR accuracy (AAR), RMSD, and docking performance. Higher values of AAR, DockQ, and SR indicate better performance, while lower RMSD, iRMS, and LRMS are preferable. DockQ scores are calculated by comparing the designed structures to the corresponding wild-type complexes.

| Method | CDR1 ↑ | CDR2 ↑ | CDR3 ↑ | DockQ ↑ | iRMS ↓ | LRMS ↓ | Success Rate ↑ |
|---|---|---|---|---|---|---|---|
| DiffAb (AF3) | 0.533 | 0.291 | 0.156 | 0.211 | 13.265 | 35.805 | 0.346 |
| IgGM | **0.565** | 0.330 | 0.183 | **0.267** | 6.927 | 14.966 | 0.415 |
| FlexRibbon | 0.500 | **0.441** | **0.218** | 0.244 | **5.571** | **13.506** | **0.437** |

## D.4 PROTEIN-LIGAND DOCKING PREDICTION

**Datasets.** Benchmarking performance without finetuning is a key way to assess a model's generalization ability. For this task, we directly evaluate our model in the pretraining setting. The test set is PoseBusters V1 (Buttenschoen et al., 2024), which contains 428 protein-ligand complexes deposited in the PDB between January 1, 2021 and May 30, 2023. For pretraining, we follow the same protocol as Boltz-1 and Protenix, using all PDB structures released before 2021-09-30. Since these three methods share same cutoff time, comparisons on test set remain fair.

**Training and inference.** Evaluating direct inference provides insight into a model's ability to directly predict complex structures from minimal inputs. Given protein sequences and ligand SMILES strings, we generate full protein-ligand complex structures in a manner similar to AlphaFold 3. For FlexRibbon, Protenix, and Boltz-1, we generate 5 samples per case using a single random seed.

Table 12: Success Rate (SR) comparison of different methods.

| Method | Success Rate (%) ↑ |
|---|---|
| **MSA-based** | |
| AlphaFold3 2019 | 76.40 |
| Chai-1 | 77.00 |
| Protenix top1 | 77.10 |
| Protenix oracle | 81.31 |
| Boltz-1 top1 | 69.62 |
| Boltz-1 oracle | 74.06 |
| **Single-sequence-based** | |
| DiffDock | 37.90 |
| DiffDock-L | 50.00 |
| FlexRibbon random1 | 71.82 |
| FlexRibbon oracle | 78.70 |

**Evaluation metrics.** Standardized metrics are critical to ensure reliable comparison across methods. Following the AlphaFold 3 protocol, we report the success rate, defined as the percentage of predictions with a pocket-aligned RMSD <2 Å. The pocket-alignment procedure is consistent with AlphaFold 3: the pocket is defined as all heavy atoms within 10 Å of any ligand heavy atom, restricted to the primary polymer chain or modified residue of the ligand, and further limited to protein backbone atoms. Baselines include MSA-based methods (AlphaFold 3, Chai-1, Protenix, Boltz-1) and single-sequence methods (DiffDock, DiffDock-L). For Protenix and Boltz-1, results are reported using the top-ranked sample out of five diffusion-generated predictions. For FlexRibbon, which does not include a confidence head, we report both the *random-1* score (performance of a randomly chosen sample) and the *oracle* score (the best of five samples selected against the ground truth). Note that Boltz-1 failed on `7M31_TDR` and `7SUC_COM` due to residue number restrictions.

## D.5 LIGAND-INDUCED CONFORMATIONAL CHANGE PREDICTION

**Datasets.** Benchmarking performance without finetuning provides insights into a model's ability to generalize without task-specific adaptation. For this task, we do not perform finetuning and directly evaluate the capability of our model. The test set, originally derived from Saldaño et al. (2022), consists of 90 apo-holo protein pairs, which were relesed between June 1994 and February 2021.

**Training and inference.** Direct inference allows us to assess the model's structural prediction ability under different input conditions. Given protein sequences and ligand SMILES strings, we generate structural predictions without any fine-tuning. Specifically, we produce five predictions for each case without ligands (apo) and five predictions with ligands obtained from the original holo complexes in the PDB. For fair comparison, we report two evaluation settings: (1) all five apo samples, and (2) a mixed set of three apo samples and two holo samples.

Table 13: Evaluation of ligand-induced conformation changes: (1) Pearson correlation ($r$) between sampled and ground-truth diversity as measured by the residue flexibility (ResFlex, absolute deviation after alignment), and (2) the ensemble TM-score (TM-ens). For residue flexibility, both global (gl.) correlations and mean/median per-target (pt.) correlations are reported; for TM-ens, mean/median correlations are reported. Higher values indicate better performance. Methods marked with $\dagger$ use MSA as input.

| Method | ResFlex $r$ (gl.) ↑ | ResFlex $r$ (pt.) ↑ | TM-ens ↑ |
|---|---|---|---|
| **Benchmark results compared against baselines** | | | |
| AlphaFlow$^\dagger$ | 0.455 | 0.527/0.527 | 0.864/0.893 |
| ESMFlow | 0.416 | 0.496/0.522 | 0.856/0.893 |
| ESMDiff | 0.424 | 0.502/0.517 | 0.851/0.883 |
| FlexRibbon | 0.503 | 0.551/0.542 | 0.889/0.920 |
| FlexRibbon with ligand | **0.519** | **0.587/0.615** | **0.901/0.931** |
| **Ablation study** | | | |
| FlexRibbon 5apo | 0.503 | 0.551/0.542 | 0.889/0.920 |
| FlexRibbon 5holo | 0.454 | 0.524/0.540 | 0.888/0.918 |
| FlexRibbon 3apo+2holo | 0.519 | 0.587/0.615 | 0.901/0.931 |
| FlexRibbon 5apo+5holo | 0.535 | 0.620/0.638 | 0.907/0.936 |

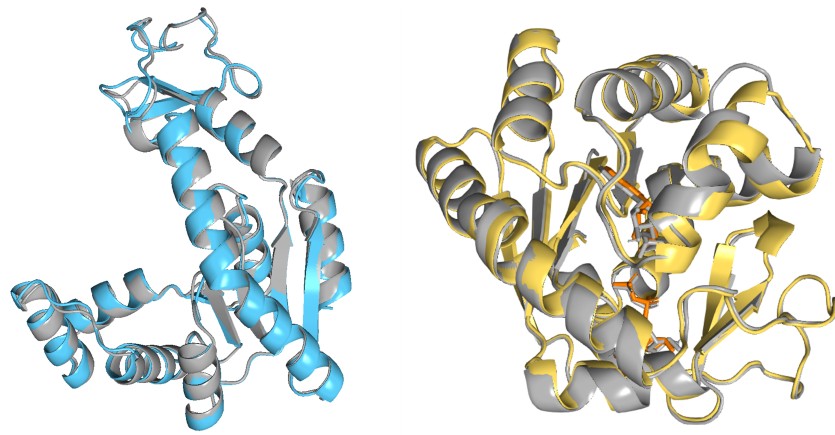

Figure 8: Overlay of the prediction of the (left) apo and (right) holo states of adenylate kinase, overlaid with the native(grey) from the PDB. The holo state includes an AMP and an ADP molecule (orange), whose presence induces the kinase to fold inwards to hold the molecules in place.

**Evaluation metrics.** Rigorous evaluation metrics are essential to capture both accuracy and diversity in structural predictions. Following the protocol in AlphaFlow (Jing et al., 2024), we use two types of metrics. The first is the Pearson correlation ($r$) between sampled diversity and ground-truth diversity, measured by residue flexibility (ResFlex, absolute deviation after alignment), reported as global (gl.) mean and per-target (pt.) mean/median correlations. The second is the ensemble TM-score (TM-ens), reported as mean and median. Results are presented in Section 4.2. In addition, we conduct an ablation study with different ligand conditions, showing FlexRibbon can generate structures in multiple conformational states, highlighting its potential to address multi-state problem.

### D.6 PROTEIN-LIGAND BINDING AFFINITY PREDICTION

**Datasets.** Reliable benchmarking requires consistent training and evaluation protocols. Following the strategy of SPIN (Choi et al., 2024), we use the same training and test sets. The training data

is drawn from PDBbind v2020 (Liu et al., 2017), comprising 19,443 protein-ligand complexes. For evaluation, we adopt the CASF-2016 (Su et al., 2018) benchmark, which includes 285 samples. To prevent data leakage, any overlapping entries between CASF-2016 and the training set were removed.

**Training and inference.** Model training is formulated as a regression task to predict binding affinity values. The input is the three-dimensional structure of protein-ligand complexes, and the target output is a continuous affinity score. During inference, the model predicts one affinity score per sample, which is directly compared against the ground-truth value.

**Evaluation metrics.** Standard regression metrics are used to assess predictive accuracy and correlation with experimental data. Specifically, we report the Root Mean Square Error (RMSE) and Pearson's correlation coefficient ($R$). Detailed results are presented in Section 4.2.

### D.7 EC NUMBER PREDICTION

**Datasets.** Accurate enzyme function prediction requires high-quality annotation datasets. We follow the dataset setup used in DeepFRI (Gligorijević et al., 2021), where enzyme annotations are derived from UniProtKB with experimentally validated Enzyme Commission (EC) numbers. The training, validation, and test set contains 15551, 1729, 1919 protein samples respectively.

**Training and inference.** The task is formulated as a multi-label classification problem, where each protein sequence may be associated with one or more EC numbers. During inference, the model outputs probability scores over all possible EC labels, which is then used to compute the precision-recall curve. For both training and inference, the protein sequences are passed through our base model once for structure prediction, after which both sequence and structural information are used for model finetuning and evaluation.

**Evaluation metrics.** Model performance is evaluated using the maximum F-score ($F_1$), which balances precision and recall. Specifically, $F_1$ is defined as the maximum F-score across all probability thresholds:

$$F_1 = \max_{t \in [0,1]} \frac{2 \cdot \text{Precision}(t) \cdot \text{Recall}(t)}{\text{Precision}(t) + \text{Recall}(t)},$$

where $t$ is the threshold applied to predicted probabilities.

### D.8 GO TERM PREDICTION

**Datasets.** Gene Ontology (GO) provides a comprehensive representation of protein function, covering three sub-ontologies: Molecular Function (MF), Biological Process (BP), and Cellular Component (CC). Following DeepFRI (Gligorijević et al., 2021), we construct the training, validation, and test set as shown in Table 14.

Table 14: Size of data samples used for the Gene Ontology (GO) task.

| Sub-ontology | Training samples | Validation samples | Test samples |
|---|---|---|---|
| BP | 23,514 | 2,624 | 3,415 |
| MF | 24,952 | 2,747 | 3,415 |
| CC | 11,298 | 1,299 | 3,415 |

**Training and inference.** GO term prediction is also framed as a multi-label classification problem. For each protein, the model outputs probability scores over GO terms independently for MF, BP, and CC. The training and inference procedures are identical to those used for EC number prediction, including structure prediction.

**Evaluation metrics.** Performance is measured by $F_1$, defined as the maximum F1-score across thresholds. The metric captures the balance between precision and recall in predicting GO terms and is widely adopted in functional annotation benchmarks.

### D.9 PROTEIN STRUCTURE PREDICTION

**Datasets.** For the protein structure prediction task, we do not perform any finetuning and therefore do not use a training set. To evaluate the performance of FlexRibbon, we adopt two standard benchmarks: CASP15 (Kryshtafovych et al., 2023). For CASP15, we obtain the data from the official website (https://predictioncenter.org/). The organizers released 45 native-domain structures derived from 30 full-chain targets, with the first target made available on May 2, 2022.

**Training and inference.** For the protein structure prediction task, we directly use the pretrained model without any task-specific finetuning. During inference, given a full-chain amino acid sequence, the model predicts atomic-level coordinates for the entire structure. Following the prediction protocol of AlphaFold 3, we generate 25 samples for each target using different random seeds and select the prediction with the highest confidence score as the final output.

**Evaluation metrics.** For CASP15, we follow the official evaluation protocol: structures are predicted at the whole-chain level and evaluated on the released domains. The primary metrics are TM-score (Zhang & Skolnick, 2004) and lDDT (Mariani et al., 2013). Baselines include AlphaFold 3, ESMFold and ESM3; we run both methods locally and evaluate all predictions using the `TMscore` program (version 20220227) and `lddt` program (version 2.11.1). As shown in Table 15, FlexRibbon achieves state-of-the-art performance among single-sequence-based methods.

Table 15: Evaluation of protein structure prediction on CASP15. Results for AlphaFold 3, ESMFold, ESM3, and FlexRibbon are obtained by running all methods locally and evaluating them using the same pipeline based on the `TMscore` and `lddt` programs. Each target is predicted at the whole-chain level and evaluated on the released domains. Methods marked with [†] use MSA as input.

| Method | TM-score ↑ | lDDT ↑ |
|---|---|---|
| AlphaFold 3[†] | 0.820 | 0.840 |
| ESMFold | 0.684 | 0.718 |
| ESM3 | 0.673 | 0.689 |
| FlexRibbon | 0.703 | 0.726 |

### D.10 FINETUNING TIME AND MEMORY

To assess the practicality of FlexRibbon for real-world downstream applications, we report the finetuning runtime and GPU memory usage for each task, as different downstream tasks vary substantially in dataset size and training configuration. The number of GPUs used and the total finetuning time are summarized in Table 16. Since the protein-antibody/nanobody interface prediction task and the antibody/nanobody design task share the same training dataset, we report their finetuning cost as a single combined entry. Overall, FlexRibbon can typically be finetuned for a wide range of real-world applications using only 8 GPUs, demonstrating both its efficiency and practical applicability.

Table 16: Finetuning cost for each downstream task. All experiments are conducted on the nodes equipped with 8 NVIDIA A100 80 GB GPUs.

| Task | #{A100 80G} | Time (hours) |
|---|---|---|
| Antibody/Nanobody interface prediction and design | 16 | 69 |
| Protein-peptide interface prediction | 8 | 15 |
| Protein-ligand docking prediction | N/A | N/A |
| Ligand-induced conformational change prediction | N/A | N/A |
| Protein-ligand binding affinity prediction | 8 | 5 |
| EC number prediction | 8 | 10 |
| GO term prediction | 8 | 10 |

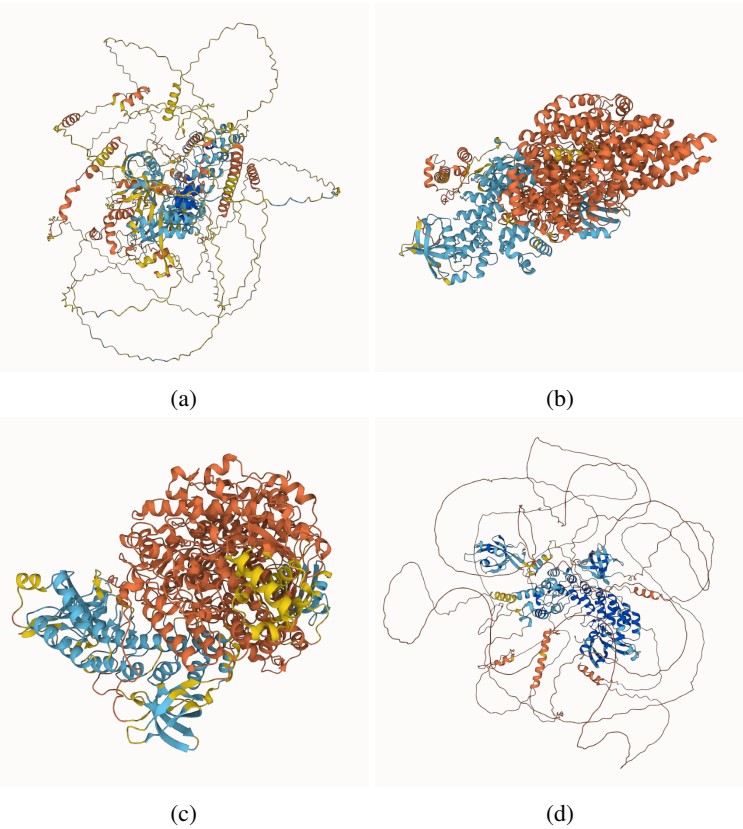

Figure 9: Comparison of disordered-region predictions for the DP02376 protein from the CAID 2 (Critical Assessment of protein Intrinsic Disorder) dataset, corresponding to the example shown in Extended Data Fig. 1 of the AlphaFold 3 paper. Predictions are shown for: (a) FlexRibbon, (b) Boltz-1, (c) Boltz-2, and (d) AlphaFold 3.

### D.11 DISORDERED REGION PREDICTION

To assess whether our model can qualitatively handle intrinsically disordered segments, we reproduced the disordered-region prediction example from AlphaFold 3's paper (Extended Data Fig. 1). The protein is DP02376 from the CAID 2 (Critical Assessment of protein Intrinsic Disorder prediction) set. The sequence contains a well-characterized disordered stretch, which AlphaFold 3 highlighted as a case where conventional structure predictors tend to over-compact flexible regions.

Figure 9 presents the same sequence predicted by FlexRibbon, Boltz-1(Wohlwend et al., 2025), Boltz-2(Passaro et al., 2025), and AlphaFold 3(Abramson et al., 2024). Consistent with AlphaFold 3's original observation, only FlexRibbon and AlphaFold 3 produce a conformation that preserves the expected disorder—reflected as a diffuse, low-confidence geometry rather than an artificially rigid fold. The remaining models collapse the flexible segment into a compact structure, failing to reflect intrinsic disorder. This qualitative reproduction demonstrates that our model captures structural uncertainty in a manner comparable to AlphaFold 3 for this illustrative case.

## E  USAGE OF LLM

We employed GPT-5 to assist in refining the writing of this manuscript. Specifically, GPT-5 was used to polish grammar, improve readability, and streamline phrasing, while all scientific content, experimental design, and data analysis were developed and verified by the authors. The use of GPT-5 was limited to language refinement and did not influence the technical contributions or conclusions of this work.

