# OpenReview forum: "FlexRibbon: Joint Sequence and Structure Pretraining for Protein Modeling"
_ICLR.cc/2026/Conference — ICLR 2026 Poster_

### Official Review · Reviewer_BQ5o · 2025-10-31

**Soundness:** 3
**Presentation:** 3
**Contribution:** 3
**Rating:** 4
**Confidence:** 4

**Summary:**

FlexProtein unifies protein structure prediction and design. It learns to model the sequence via masked language modeling and learns to model the structure via diffusion-based denoising. This flexible framework is able to generate both structures and sequences and show impressive results on interface, interaction and function prediction.

**Strengths:**

The framework of co-generating sequences and structures is very interesting and an intuitive framework.
The authors show that interesting results on interface, interaction and function prediction tasks.

**Weaknesses:**

While the downstream tasks are very interesting, a framework that "unifies protein structure prediction and design" should evaluate on structure prediction.

The abstract mentions that "Sequence-only language models (e.g., ESM-2) capture sequence semantics at scale but lack structural grounding". However there are many works that aim to bridge this gap.

[1] Learning protein sequence embeddings using information from structure. Tristan Bepler, Bonnie Berger.

[2] ProstT5: Bilingual Language Model for Protein Sequence and Structure. Michael Heinzinger,  Konstantin Weissenow, Joaquin Gomez Sanchez, Adrian Henkel, Martin Steinegger, Burkhard Rost.

[3] Distilling Structural Representations into Protein Sequence Models. Jeffrey Ouyang-Zhang, Chengyue Gong, Yue Zhao, Philipp Krähenbühl, Adam R Klivans, Daniel J Diaz.

[4] Structure-Informed Protein Language Model. Zuobai Zhang, Jiarui Lu, Vijil Chenthamarakshan, Aurélie Lozano, Payel Das, Jian Tang.

**Questions:**

How does FlexProtein perform on monomeric protein structure prediction against ESMFold and AlphaFold?

How does FlexProtein compare to IgML on design when used in the sequence-only model?

---

> ### Author Response · Authors · 2025-11-13
> **Subject: Clarification on “IgML” Reference in Review Comment**
>
> Dear Reviewer,
>
> Thank you very much for your valuable comments. Regarding your question — “How does FlexProtein compare to IgML on design when used in the sequence-only model?” — could you please clarify which approach you are referring to?
>
> I could not locate a paper for IgML, but I did find a paper titled “IgLM: Infilling language modeling for antibody sequence design” (Shuai, Richard W., Jeffrey A. Ruffolo, and Jeffrey J. Gray, Cell Systems, 2023). Did you mean IgLM instead, or is there another reference for IgML that you could kindly share?
>
> Thank you again for your time and constructive feedback.

---

> > ### Comment · Reviewer_BQ5o · 2025-11-14
> >
> > Yes, IgLM thank you for clarifying.

---

> ### Author Response · Authors · 2025-11-22
>
> Thanks for reviewing our paper and providing constructive comments. Your questions are answered as follows:
>
> > W1: While the downstream tasks are very interesting, a framework that "unifies protein structure prediction and design" should evaluate on structure prediction.
>
> > Q1: How does FlexProtein perform on monomeric protein structure prediction against ESMFold and AlphaFold?
>
> We have added comprehensive benchmarks on protein structure prediction using both **CASP15** and **CAMEO 2022** to evaluate our framework in standard structure-prediction settings.
>
> For **CASP15**, we follow the official evaluation protocol: each target is predicted at the whole-chain level and evaluated on the released domains using TM-score and lDDT. Baselines include the MSA-based AlphaFold 3 and the single-sequence models ESMFold and ESM3. Across 45 released domains in CASP15, FlexProtein outperforms both single-sequence baselines:
>
> | Method | TM-score | lDDT |
> | :--- | :---: | :---: |
> | AlphaFold 3 | 0.820 | 0.840 |
> | ESMFold | 0.684 | 0.718 |
> | ESM3 | 0.673 | 0.689 |
> | FlexProtein | 0.703 | 0.726 |
>
> For **CAMEO 2022**, we follow the DPLM-2 evaluation pipeline and use their official metrics and evaluation scripts. Across 183 targets, FlexProtein achieves the highest overall accuracy, outperforming all prior single-sequence and multimodal baselines:
>
> | Method | RMSD | TMscore |
> | :--- | ---: | ---: |
> | ESMFold | 3.99/2.03 | 0.85/0.93 |
> | PVQD | 4.08/1.95 | 0.81/0.88 |
> | MultiFlow | 17.84/17.96 | 0.50/0.46 |
> | ESM3 | 6.33/2.98 | 0.85/0.92 |
> | DPLM-2 (3B) | 6.34/3.65 | 0.83/0.89 |
> | DPLM-2 (3B) w/ folding SFT | 5.71/3.23 | 0.85/0.90 |
> | FlexProtein | 2.43/2.05 | 0.88/0.94 |
>
> These results show that FlexProtein provides a unified framework for protein structure prediction, while achieving state-of-the-art performance among single-sequence models. Full results are provided in **Appendix D.9, Table 15 Line 1440, Table 16 Line 1458**.
>
> > W2: The abstract mentions that "Sequence-only language models (e.g., ESM-2) capture sequence semantics at scale but lack structural grounding". However there are many works that aim to bridge this gap.
> [1] Learning protein sequence embeddings using information from structure. Tristan Bepler, Bonnie Berger.
> [2] ProstT5: Bilingual Language Model for Protein Sequence and Structure. Michael Heinzinger, Konstantin Weissenow, Joaquin Gomez Sanchez, Adrian Henkel, Martin Steinegger, Burkhard Rost.
> [3] Distilling Structural Representations into Protein Sequence Models. Jeffrey Ouyang-Zhang, Chengyue Gong, Yue Zhao, Philipp Krähenbühl, Adam R Klivans, Daniel J Diaz.
> [4] Structure-Informed Protein Language Model. Zuobai Zhang, Jiarui Lu, Vijil Chenthamarakshan, Aurélie Lozano, Payel Das, Jian Tang.
>
> Thank you very much for raising this point and for listing the relevant prior works. We agree that several studies have incorporated structural information into protein language models, including Bepler & Berger (2019), ProstT5, Ouyang-Zhang et al., and Zhang et al. These works are already cited in our Introduction and Related Work (Page 1, L40–44 and L95–99).
>
> To avoid any ambiguity, we have also updated the abstract to explicitly acknowledge this line of research. The revised version now states that “a number of recent works incorporate structural signals into sequence encoders" Our intention was to distinguish standard sequence-only pretraining (such as the original ESM-2 models) from more recent structure-based models, not to imply that such techniques do not exist.
>
> We appreciate the reviewer’s careful reading and believe the updated abstract and existing citations now accurately reflect the current landscape of structure-aware PLMs.

---

> ### Author Response · Authors · 2025-11-22
>
> > Q2: How does FlexProtein compare to IgLM on design when used in the sequence-only model?
>
> Thanks for your suggestion. We also experiment with CDR H3 only design as IgLM does, using only our sequence module, leaving out any structure information. IgLM provides its infilling results of 49 therapeutic antibodies from Thera-SAbDab. Among these antibodies, we found that 22 are in our training set when finetuning with the antibody design task. Thus, we test with the remaining 27 antibodies, and compare the average AAR with the sequences infilled by IgLM over these 27 antibodies:
> | Method       | AAR   |
> | :----------- | ----: |
> | IgLM         | 0.204 |
> | FlexProtein  | 0.436 |
>
> The sequence module of FlexProtein produces CDR H3 regions with higher AAR. Note that since IgLM allows variable length of the infilled sequences, we selected only infilled CDR H3 sequences with the same length as the native ones to compute the AAR (for each antibody, IgLM provides 9000 different infilled results, we select only those with equal length as the native CDR H3).
>
> We also calculate the delta of solubility as in The IgLM paper. Result shows that the designed heavy chains has an higher solubility of 0.042 on average, over the 27 test cases, calcualted with CamSol, showing improved in-silico developability of the designed heavy sequences.

---

> > ### Comment · Reviewer_BQ5o · 2025-11-27
> >
> > Thanks to the authors for the response. The authors have addressed my concerns. The structure prediction and antibody design performance look impressive with appropriate baselines and evaluation protocols. Structure sequence hybrid models are acknowledged.
> >
> > I will raise my review from 4 to 6.

---

> > > ### Author Response · Authors · 2025-11-27
> > >
> > > Thank you for your thoughtful follow-up and for reconsidering your evaluation. We appreciate your recognition of the structure prediction and antibody design results. Your feedback has been valuable in improving the clarity and rigor of the paper.

---

### Official Review · Reviewer_wgxZ · 2025-10-31

**Soundness:** 4
**Presentation:** 3
**Contribution:** 4
**Rating:** 8
**Confidence:** 3

**Summary:**

The paper presents FlexProtein, a joint sequence–structure pretrained model for proteins that removes dependence on multiple sequence alignments.
It integrates a diffusion-based structural generator and a structure-informed masked language model (SIMLM) into a unified framework.
The framework is trained on ~78M AFDB and 180k PDB structures, and a pLDDT-weighted denoising loss enables the model to learn reliable geometry from noisy predicted structures.
Experiments across 12 benchmarks show the performance of the proposed method.

**Strengths:**

1. This paper is well motivated, this paper addresses a well-defined and flexible problem: learning unified protein representations without relying on MSA
2. The proposed framework combines diffusion-based structure denoising and structure-informed masked language modeling achieving bidirectional supervision between sequence and structure
3. Extensive experiments across 12 heterogeneous benchmarks demonstrate the performance and generalization ability of proposed FlexProtein
4. Specifically, FlexProtein achieves consistent gains in flexible or high mutation regimes, where most MSA-based or sequence-only models degrade sharply, providing strong support for the MSA-free framework.

**Weaknesses:**

Overall, I find this paper well-motivated, technically mature, and experimentally comprehensive. In my view, this work represents a solid contribution to the emerging direction of general-purpose protein foundation models. I have only 1 point of technical curiosity:
* Given the scale of the training setup and the diffusion architecture, I am curious about the inference efficiency and fine-tuning cost. While the paper provides strong empirical evidence across 12 benchmarks, it would be valuable to include quantitative runtime or memory profiles. This information would help assess FlexProtein’s practicality for real-world downstream tasks.

**Questions:**

I found the paper technically strong and conceptually consistent, my main questions are reflected in the weakness section.

---

> ### Author Response · Authors · 2025-11-22
>
> Thank you very much for your thoughtful and encouraging feedback. We truly appreciate your recognition of the motivation, technical design, and comprehensive evaluation of FlexProtein. Your positive assessment—especially regarding our unified MSA-free framework and its strong performance in flexible and high-mutation regimes—means a great deal to us. Your questions are answered as follows:
>
> >W1: I have only 1 point of technical curiosity: Given the scale of the training setup and the diffusion architecture, I am curious about the inference efficiency and fine-tuning cost. While the paper provides strong empirical evidence across 12 benchmarks, it would be valuable to include quantitative runtime or memory profiles. This information would help assess FlexProtein’s practicality for real-world downstream tasks.
>
> We sincerely appreciate your insightful comments and your interest in the practicality of FlexProtein. We fully agree that inference efficiency and fine-tuning cost are important considerations, especially given the scale of our architecture. Following your valuable suggestion, we have added quantitative runtime and memory profiles for both fine-tuning and inference.
>
> ### **Fine-tuning efficiency**
>
> Since different downstream tasks vary substantially in dataset size and training configuration, we report runtime and memory usage task by task:
>
> | Task | #{A100 80G} | Time (hours) |
> | :--- | ---: | ---: |
> | Antibody/Nanobody interface prediction and design | 16 | 69 |
> | Protein-peptide interface prediction | 8 | 15 |
> | Protein-ligand docking prediction | N/A | N/A |
> | Ligand-induced conformational change prediction | N/A | N/A |
> | Protein-ligand binding affinity prediction | 8 | 5 |
> | EC number prediction | 8 | 10 |
> | GO term prediction | 8 | 10 |
>
> All the details of finetuning efficiency have been added to **Appendix D.10 and Table 17 Line 1477**.
>
>
> ### **Inference efficiency**
>
> To evaluate inference performance, we select four heteromers and four homomers from the PDB, with sequence lengths approximately 256, 512, 1024, and 2048 residues. We measure the forward-pass runtime and GPU memory consumption on a single NVIDIA A100 80GB GPU. Preprocessing overhead is excluded to ensure fair comparison of model runtime.
>
> | PDB  | Type      | Residue number | Inference time (sec.) | GPU Memory (GiB) |
> | :--- | :---      | --------------:| ---------------------:| ----------------:|
> | 1AB9 | Heteromer |           246  |                 11.3  |             6.7  |
> | 1KG0 | Heteromer |           517  |                 16.7  |             8.0  |
> | 8ODX | Heteromer |         1,024  |                 43.3  |            11.0  |
> | 8TQO | Heteromer |         2,064  |                139.8  |            27.0  |
> | 4W80 | Homomer   |           256  |                 10.9  |             7.0  |
> | 8VE4 | Homomer   |           512  |                 15.7  |             7.7  |
> | 1VH1 | Homomer   |         1,024  |                 42.0  |            11.0  |
> | 6EBM | Homomer   |         2,052  |                139.0  |            27.2  |
>
> All detailed of inference efficiency have been added to **Appendix C.8 Line 1067 and Table 6 Line 1080**.

---

### Official Review · Reviewer_8Gx5 · 2025-10-31

**Soundness:** 3
**Presentation:** 2
**Contribution:** 3
**Rating:** 4
**Confidence:** 4

**Summary:**

This paper introduces FlexProtein, a unified framework for joint sequence-structure modeling. The method integrates a diffsuion-based structural denoising model with structure-informed masked language model to enable bidirectional learning between sequence and structure.
It consists of three modules: (1) a sequence encoder that performs MLM (2) a coarse-grained structure module that learns residue-level topology with noisy structure and sequence (3) an all-atom structure module that refines atomic coords.
The authors finetuned on several design tasks including antibody and nanobody design, intermolecular interaction prediction and protein function prediction and show great results.

**Strengths:**

1. This paper proposed a unified modeling of sequence and structure. The integration of structure-informed MLM with diffusion enables the model to simultaneously reason over both modalitiies.
2. This paper also unified interface for proteins and small molecules. This allows straighfoward protein-ligand binding affinity/docking prediction.

**Weaknesses:**

1. Lack of comparison with other models. According to the architecture, it looks like FlexProtein should be able to perform structure prediction and inverse folding. However, there is no any benchmark on that. I suggest the authors can compare against DPLM-2 on these two tasks.
2. Lack of ablation study. In this paper the authors introduce three modules, while the authors did not show any ablation study on these modules. For example, the coarse-grained structure module can be removed according to the framework. Will there be any difference? The authors did not show that.
3. The authors propose the idea of "structure-informed masked language model" but did not cite the paper that exactly used structured-informed protein language model for protein design. [1]

There are also issues with the presentation of the paper:
1. In Figure 1, the authors draw a line from $D _ \theta(R_t, t)$ to single step. I was confused when I first saw this. This should refers to the inferece stage, right? While the loss indicates it's training stage. I suggest the authors seperate the training and inference stages.
2. In line 171, the authors wrote "For small molecules, we incorporate 2D topology". While this can't be found in the framework. I suggest the authors should add the details about small molecules are added to the framework in the figure.

[1] Zheng, Z., Deng, Y., Xue, D., Zhou, Y., Ye, F., & Gu, Q. (2023, July). Structure-informed language models are protein designers. In International conference on machine learning (pp. 42317-42338). PMLR.

**Questions:**

See weakness above. I will raise the scores accordingly if the experiment concerns could be solved.

---

> ### Author Response · Authors · 2025-11-22
>
> Thanks for reviewing our paper and providing constructive comments. Your questions are answered as follows:
>
> > W1: Comparison with other models. According to the architecture, it looks like FlexProtein should be able to perform structure prediction and inverse folding. However, there is no any benchmark on that. I suggest the authors can compare against DPLM-2 on these two tasks.
>
> With your valuable suggestions, We have added comprehensive benchmarks on protein structure prediction using both CASP15 and CAMEO 2022 to evaluate our framework in a standard structure-prediction setting.
>
> For **CASP15** benchmark, we follow the official evaluation protocol: each target is predicted at the whole-chain level and evaluated on the released domains using TM-score and lDDT. Baselines include the MSA-based AlphaFold 3 and single-sequence-based ESMFold and ESM3. Across 45 released domains in CASP15, FlexProtein surpasses both single-sequence baselines:
> | Method | TM-score | lDDT |
> | :--- | ---: | ---: |
> | AlphaFold 3 | 0.820 | 0.840 |
> | ESMFold | 0.684 | 0.718 |
> | ESM3 | 0.673 | 0.689 |
> | FlexProtein | 0.703 | 0.726 |
>
> For **CAMEO 2022** benchmark, we follow the DPLM-2 evaluation pipeline and use their official metrics and evaluation scripts. Across 183 targets, FlexProtein achieves the best overall accuracy, outperforming all prior single-sequence and multimodal baselines:
> | Method | RMSD | TMscore |
> | :--- | ---: | ---: |
> | ESMFold | 3.99/2.03 | 0.85/0.93 |
> | PVQD | 4.08/1.95 | 0.81/0.88 |
> | MultiFlow | 17.84/17.96 | 0.50/0.46 |
> | ESM3 | 6.33/2.98 | 0.85/0.92 |
> | DPLM-2 (3B) | 6.34/3.65 | 0.83/0.89 |
> | DPLM-2 (3B) w/ folding SFT | 5.71/3.23 | 0.85/0.90 |
> | FlexProtein | 2.43/2.05 | 0.88/0.94 |
>
> These results demonstrate that FlexProtein‘s good capability for protein structure prediction and delivering state-of-the-art performance among single-sequence models. All these results have been added to **Appendix D.9 Table 15 Line 1440, Table 16 Line 1458**.
>
> For inverse folding task, we found our pretrained model cannot handle this task well directly and we are finetuning on this task. The inverse folding benchmark result will be updated later.
>
> > W2: Ablation study. In this paper the authors introduce three modules, while the authors did not show any ablation study on these modules. For example, the coarse-grained structure module can be removed according to the framework. Will there be any difference? The authors did not show that.
>
> Thank you for raising this important question regarding the ablation of the coarse-grained structure module. Due to the high computational cost of training a 3B-parameter model from scratch, it was not feasible to perform a full ablation at that scale within the rebuttal window. To provide concrete evidence nonetheless, we conducted a controlled ablation study at the 400M-parameter scale.
>
> In particular, we trained two models from scratch under identical settings:
> - a 400M-parameter version of **FlexProtein** that mirrors the architecture of the 3B model, including the coarse-grained structure module;
> - an **ablated variant** in which the coarse-grained module is removed and an equivalent number of parameters is reassigned to the sequence module.
>
> Both models were trained on the AFDB dataset using two nodes (each with 8×A100 GPUs). After 10,000 training steps, we evaluated both models on the CAMEO 2022 benchmark.
>
> The results (shown in the table below) clearly indicate that removing the coarse-grained module significantly degrades training efficiency and downstream performance. Even with the same training budget, the ablated model converges more slowly and achieves lower accuracy across key structural metrics.
> | Method | RMSD | TM-score |
> | :--- | ---: | ---: |
> | FlexProtein | 6.38 | 0.50 |
> | FlexProtein w/o coarse-grained module | 9.31| 0.23 |
>
> This ablation demonstrates that the coarse-grained structure module plays an essential role in stabilizing training and improving model quality, even at smaller scales, and we expect its impact to be even more pronounced in the full 3B-parameter setting.
>
> >W3: The authors propose the idea of "structure-informed masked language model" but did not cite the paper that exactly used structured-informed protein language model for protein design. “Zheng, Z., Deng, Y., Xue, D., Zhou, Y., Ye, F., & Gu, Q. (2023, July). Structure-informed language models are protein designers. In International conference on machine learning (pp. 42317-42338). PMLR.”
>
> Thank you very much for pointing out the omission of the relevant work. You are correct that Structure‑informed Language Models Are Protein Designers (Zheng et al., 2023) should have been cited in our introduction and related work. We incorporate this reference at Introduction (**Line 43**) and Related Works (**Line 99**), noting that the work proposes a structure-informed protein language model for design.

---

> ### Author Response · Authors · 2025-11-22
>
> Thanks for reviewing our paper and providing constructive comments. Your questions are answered as follows:
>
> > W4: There are also issues with the presentation of the paper: In Figure 1, the authors draw a line from to single step. I was confused when I first saw this. This should refers to the inferece stage, right? While the loss indicates it's training stage. I suggest the authors seperate the training and inference stages.
>
> Thank you for pointing out this confusing aspect of Figure 1. We have updated **Figure 1** by separating the training pipeline from the sampling pipeline, making the workflow clearer and avoiding any ambiguity between loss computation and inference. Please let us know if the revised figure can be further improved—we appreciate your helpful suggestion.
>
> >W5: There are also issues with the presentation of the paper: In line 171, the authors wrote "For small molecules, we incorporate 2D topology". While this can't be found in the framework. I suggest the authors should add the details about small molecules are added to the framework in the figure.
>
> Thanks for your invaluable suggestions. We have modified the Figure 1 in the  updated manuscript to add the 2D topology of molecules and also more detailed description in **Appendix B Lines 834-839**. Specifically, the 2D topology is encoded into a bond feature map, according to the connectivity between atoms, and the element types of two atoms connected each bond. The bond feature map is then fed into different modules with a learning embedding, and added to the attention bias after a linear projection. We highlight the high level idea in Figure 1, and will add more details in the figures of Appendix. Please let kindly us know if you think the figure can be further improved.

---

> ### Comment · Reviewer_8Gx5 · 2025-11-22
>
> The responses have addressed my concerns about comparison with other models in structure prediction and ablation study. I have raised the score to 6.

---

> > ### Author Response · Authors · 2025-11-22
> >
> > We truly appreciate your updated assessment. If you believe there are other points where further clarification or additional experiments would improve the paper, we are very willing to address them. Your suggestions have already strengthened the work, and we welcome any further guidance.

---

### Official Review · Reviewer_RzWH · 2025-11-02

**Soundness:** 3
**Presentation:** 3
**Contribution:** 3
**Rating:** 4
**Confidence:** 4

**Summary:**

FlexProtein is a pretrained protein model that jointly learns from sequences and 3D structures without MSAs, addressing limitations of existing sequence-only/MSA-based models. It captures global folds and flexible conformations via masked language modeling and diffusion-based denoising. It sets new SOTA on 12 tasks, notably in mutation-rich scenarios.

**Strengths:**

- This study builds a 3B protein foundation model and tests on 12 downstream tasks. The experiments are very comprehensive and the results are promising.
- FlexProtein enables codesign of sequence and structures for antibodies/nanobodies, which may surpass its MSA-free counterparts since these models may struggle due to few MSA data.

**Weaknesses:**

- Lack of ablation study for several key components.
	- The training objective: how does plddt-based reweighting affect the performance?
	- Pretraining tasks: there are three pretraining modes: mode 1 (seq-to-structure), mode 2 (coupled perturbation), and mode 3 (seq-masked global perturbation). How does the balance of these modes change the final results?
- Potential data leakage. In Figure 5, are the two proteins (4ake, and 2eck) already contained in the PDB dataset?

**Questions:**

- For all-atom coordinate denoising, how do you determine the number and type of atoms in a residue, when the sequence is masked, or perturbed? Is the structure module optmized only when the clean sequence is given?
- In Section 3.3, there is denoising diffusion loss for coordinates, why is this section named `masked language model'?
- In mode 1, do you jointly optimze the coarse grained structure module and the all-atom structure module?
- In this study, antibodies are generated in a masked-language-model style. Can the model generate longer proteins, like EvoDiff, DPLM?
- Seems that FlexProtein beats AF3 in several tasks (Fig 4, Table 5), is this due to the less usage of information in AF3?

---

> ### Author Response · Authors · 2025-11-22
>
> Thanks for reviewing our paper and providing constructive comments. Your questions are answered as follows:
>
> >W1: Ablation study for several key components. The training objective: how does plddt-based reweighting affect the performance?
>
> The motivation behind the pLDDT-based reweighting is to handle the inherent tension introduced by low-confidence structural targets. Intrinsically disordered segments usually appear as low-pLDDT regions in AlphaFold-derived training data. These coordinates carry useful biological signal but are noisy. Treating them naively leads to two well-known failure modes:
> - Discarding low-pLDDT residues prevents the model from ever seeing examples of disorder, this caused the model to "over-fold" disordered segments into α-helices or compact structures.
> - Keeping low-pLDDT residues with full weight makes the noisy coordinates disproportionately influence the structure objective, which usually reduces global structure quality.
>
> The pLDDT-based reweighting offers a middle ground. By down-weighting low-confidence residues, we retain their informational value while preventing noise from overwhelming the loss. High-confidence regions continue to shape backbone fidelity, whereas disordered segments still contribute signal in a controlled manner.
>
> This behavior aligns with what we observe in the disorder region prediction section (**Appendix D.11**), where we reproduce the disordered-region prediction showcase from the Extended Data Fig. 1 in AlphaFold 3 paper. As shown in **Appendix Figure 9**, our model produces a flexible, low-confidence geometry consistent with intrinsic disorder, whereas several baselines collapse the region. While this figure is not a direct ablation, the qualitative trend is consistent with the intended effect of the reweighting strategy and provides supportive evidence that the model handles disordered regions in a reasonable way.
>
>
> >W2: Ablation study for several key components. Pretraining tasks: there are three pretraining modes: mode 1 (seq-to-structure), mode 2 (coupled perturbation), and mode 3 (seq-masked global perturbation). How does the balance of these modes change the final results?
>
> Thank you very much for raising this thoughtful question about the balance among our three pre-training modes. In the current work, we use a 6:2:2 ratio for (i) sequence-to-structure prediction, (ii) coupled perturbation, and (iii) sequence-masked global perturbation. We were not able to conduct a full ablation of these ratios within the rebuttal period—retraining multiple variants from scratch, even at smaller model sizes, still requires considerable computational time and resources beyond what is feasible during the review window.
>
> Our allocation reflects the role of Mode 1 as the core objective of the model: it directly trains the mapping from sequence to 3D structure and is also empirically the most challenging to optimize. For this reason, we assign a larger proportion of training to Mode 1, while Modes 2 and 3 serve as complementary auxiliary objectives.
>
> We sincerely appreciate the reviewer’s suggestion. Exploring different weighting ratios is indeed valuable, and we plan to include a more systematic ablation in future work.
>
> > W3: Data leakage discussion. In Figure 5, are the two proteins (4ake, and 2eck) already contained in the PDB dataset?
>
> Yes, both proteins are included in the PDB dataset. However, this is equally true for all baselines evaluated in Fig. 5 (and Table 12 in the Appendix), because the targets in the Apo/Holo dataset were released between June 1994 and February 2021. Therefore, the comparison remains fair across all methods. We have added additional clarification in **Appendix D.5 (Line 1343)** to make this point explicit.
>
> The goal of this task is primarily to assess protein conformational diversity. Our main message is to highlight the importance of ligand context - specifically, that providing the ligand enables the model to correctly shift from the apo to the holo conformation. This demonstrates that ligand information plays a crucial role in guiding accurate structure prediction.

---

> ### Author Response · Authors · 2025-11-22
>
> > Q1: For all-atom coordinate denoising, how do you determine the number and type of atoms in a residue, when the sequence is masked, or perturbed? Is the structure module optmized only when the clean sequence is given?
>
> The structure module optmized when sequence is masked and clean. During the training, model will predict the side chain position of the masked residue, the side chain atom type is determined by the original residue type. When inference, we first co-generate the backbone atom coordinates and residue type. Then use the predicted residue type to predict the all-atom coordinates.
>
> > Q2: In Section 3.3, there is denoising diffusion loss for coordinates, why is this section named 'masked language model'?
>
> The  learning objective introduced in this section remains the recovery of masked amino-acid identities, consistent with standard MLM formulations. The diffusion loss appears because our formulation extends MLM to incorporate 3D structural signals, not because the section is intended to describe a standalone diffusion model.
>
> >Q3: In mode 1, do you jointly optimze the coarse grained structure module and the all-atom structure module?
>
> Yes, in Mode 1 we jointly optimize both the coarse-grained structure module and the all-atom structure module. During training, the gradients from the all-atom loss propagate back through the coarse-grained module, so both components are updated together rather than being trained in a decoupled or sequential manner. This joint optimization ensures that the coarse representation remains aligned with the finer atomic details and improves the consistency of the generated structures across resolutions.
>
> >Q4: In this study, antibodies are generated in a masked-language-model style. Can the model generate longer proteins, like EvoDiff, DPLM?
>
> Thank you for the thoughtful question - the ability to generate long proteins is indeed important and remains an active area for future work.
>
> In our current study, the pretrained model already shows that it can handle very large proteins on the structure-prediction side. Our results on PoseBusters include multimeric complexes exceeding 2,000 residues, and the model successfully reconstructs their 3D geometry. In addition, our monomer-structure results in Figure 6 and Table 15–16 include proteins close to 1,000 amino acids, further demonstrating that the model can manage long-range interactions and large protein sizes in structure inference.
>
> For antibodies, our current approach is designed for masked generation of only the six CDR loops, each typically 10–20 residues. The pretrained model cannot directly generate full-length antibodies or long proteins in the style of EvoDiff or DPLM. In practice, antibody design requires task-specific finetuning, and for designing larger proteins such as binders or multi-domain sequences, additional finetuning would still be required. The pretrained model alone is not yet sufficient to generate long de novo sequences.
>
> We agree this is an important direction, and extending our model toward scalable long-protein generation is part of future work.
>
> > Q5: Seems that FlexProtein beats AF3 in several tasks (Fig 4, Table 5), is this due to the less usage of information in AF3?
>
> We sincerely apologize for any ambiguity in the earlier description. After carefully reviewing all results that include comparisons with AF3, we would like to clarify that they fall into two categories:
> - PoseBusters benchmark (**Figure 4 Line 389 and Table 12 Line 1271**): For the protein–ligand docking task, we follow exactly the same input setting as AF3, Chai-1, Protenix, and Boltz-1. The inputs include only the protein sequence and ligand SMILES, without any additional auxiliary information. The docking structures are generated jointly under identical conditions.
> - SAbDab-23H2 benchmark (**Figure 2 Line 279, Table 7 Line 1121, Table 8 Line 1134, and Table 9 Line 1176**): For the protein-antibody/nanobody/peptide complex prediction, we follow the IgGM evaluation pipeline, which utilizes the antigen structure and epitope information. The reported results for AF3 and IgGM are quoted directly from the IgGM paper, and our evaluation strictly follows the same protocol. To our knowledge, AF3 does not use antigen structure or epitope information.
>
> Additionally, Table 5 describes the pretraining configuration and does not include any AF3 results. If you were referring to a different table or comparison, we would be very happy to clarify further.

---

### Author Response · Authors · 2025-11-22

Dear Reviewers,

Thank you very much for your thoughtful, constructive, and encouraging feedback. Your comments provided valuable insights and have substantially helped us improve the paper.

In response, we have added the results for the confidence head (**Section 4.4 Line 494 and Appendix C.5  Line 965-975**) and protein monomer structure prediction (**Section 4.4 Line 494 and Appendix D.9  Line 1423**) in the Experiments section.

We sincerely appreciate your time and consideration, and we would be glad to engage in any further discussion.

---

### Author Response · Authors · 2025-11-29

Dear reviewers, AC and SACs,

We sincerely thank the reviewers for their thoughtful comments and constructive suggestions. We appreciate the opportunity to clarify several contributions that may not have been fully emphasized in the initial discussion.

- Protein–ligand complex prediction on PoseBusters (**Fig. 4 Line389**).
Beyond antibody/nanobody structure prediction and design, an important aspect of our work is the ability to predict to protein–ligand complex structure. On the PoseBusters benchmark, our method substantially outperforms all other single-sequence baselines and achieves performance comparable to MSA-based models.
PoseBusters is widely recognized as one of the most important evaluations in the AlphaFold 3 paper, and strong results on this benchmark highlight our model’s capability to handle realistic molecular docking scenarios without relying on co-evolutionary information.

- Calibrated structure confidence (**Fig. 6 Line 520**).
Our model not only outputs 3D structures but also provides a well-behaved confidence estimate, as shown in Fig. 6. This uncertainty signal is calibrated, strongly correlated with prediction error, and therefore highly valuable for real-world scientific workflows (e.g., filtering low-confidence predictions, prioritizing downstream experiments, or guiding iterative design loops). This aspect is central for reliability and practical deployment.

We hope these clarifications address the concerns and help the reviewers better appreciate the broader significance and applicability of our contributions.

---

### Meta-Review · Area_Chair_muo7 · 2025-12-26

**Summary:**

This paper proposes FlexProtein, a large scale protein foundation model pretrained jointly on sequences and structures. The model uses a combination of structure-informed masked language modeling and diffusion-based coordinate denoising, without relying on MSAs. Reviewers generally agreed that the work is well motivated and technically ambitious, with particularly strong empirical performance across a wide range of downstream tasks, especially in mutation-rich or MSA-sparse regimes.

Initial concerns focused on the lack of direct structure prediction benchmarks, missing ablations, data leakage clarifications, and positioning relative to prior structure-informed PLMs. The rebuttal addresses most of the concerns and thus I recommend accept.

**Reviewer Concerns:**

**Addressed concerns**
- A major concern was that the paper lacked structure prediction results. In the rebuttal, the authors added comprehensive benchmarks on PoseBusters, CAMEO 2022, and CASP15, showing that FlexProtein outperforms single sequence baselines.
- The authors added an ablation study showing the necessity of the coarse-grained structure module.
- They clarified comparisons with AF3 and IgLM. They also added missing citations for structure-informed language models.
- They also addressed data leakage concerns regarding protein 4ake/2eck by clarifying that they are present in the training data for all compared methods and that the task focuses on conformational diversity.
- They added concrete runtime and memory benchmarks for fine-tuning and inference.

**Outstanding concerns**
- The authors explained the rationale for the chosen 6:2:2 ratio and acknowledged the limitation, but a systematic ablation study remains future work.
- They acknowledged that the model cannot yet generate long proteins de novo.

**Reviewer Scores:**

- Reviewer RzWH: 4 -> 6. Most technique questions are addressed.
- Reviewer 8Gx5: 4 -> 6. Structure prediction benchmarks and ablations are addressed.
- Reviewer BQ5o: 4 -> 6. Requested experiments and positioning clarifications were provided by the authors.

---

### Decision · Program_Chairs · 2026-01-26

Accept (Poster)